# Mixing Expert Knowledge:
# Bring Human Thoughts Back To the Game of Go

**Yichuan Ma[1,2], Linyang Li[1†], Yongkang Chen[1], Peiji Li[1,2], Jiasheng Ye[2],**
**Qipeng Guo[1], Dahua Lin[1], Kai Chen[1],**
[1]Shanghai AI Laboratory, Shanghai,
[2]School of Computer Science, Fudan University, Shanghai
yichuanma24@m.fudan.edu.cn, lilinyang@pjlab.org.cn

## Abstract

Large language models (LLMs) have demonstrated exceptional performance in
reasoning tasks such as mathematics and coding, matching or surpassing human
capabilities. However, these impressive reasoning abilities face significant chal-
lenges in specialized domains. Taking Go as an example, although AlphaGo has
established the high performance ceiling of AI systems in Go, mainstream LLMs
still struggle to reach even beginner-level proficiency, let alone perform natural
language reasoning. This performance gap between general-purpose LLMs and
domain experts is significantly limiting the application of LLMs on a wider range
of domain-specific tasks. In this work, we aim to bridge the divide between LLMs'
general reasoning capabilities and expert knowledge in domain-specific tasks. We
perform mixed fine-tuning with structured Go expertise and general long Chain-of-
Thought (CoT) reasoning data as a cold start, followed by reinforcement learning to
integrate expert knowledge in Go with general reasoning capabilities. Through this
methodology, we present **LoGos**, a powerful LLM that not only maintains outstand-
ing general reasoning abilities, but also conducts Go gameplay in natural language,
demonstrating effective strategic reasoning and accurate next-move prediction. Lo-
Gos achieves performance comparable to human professional players, substantially
surpassing all existing LLMs. Through this work, we aim to contribute insights
on applying general LLM reasoning capabilities to specialized domains. We will
release the first large-scale Go dataset for LLM training, the first LLM Go evalua-
tion benchmark, and the first general LLM that reaches human professional-level
performance in Go at: `https://github.com/Entarochuan/LoGos`.

## 1   Introduction

Large language models (LLMs) exemplified by OpenAI-o1 and DeepSeek-R1 [Jaech et al., 2024,
Guo et al., 2025] have demonstrated remarkable performance on reasoning tasks such as mathematics
and coding, reaching or surpassing human expert capabilities. However, when focusing on specialized
domains with scarce corpora, LLMs reach boundaries in their reasoning capabilities. In the case of
the Go game, back in 2016, AlphaGo [Silver et al., 2016] has already demonstrated the extraordinary
potential of AI systems. AlphaZero [Silver et al., 2017] further challenged human intelligence by
establishing that AI systems could "master the game of Go without human knowledge". In stark
contrast, existing general LLMs perform far below even beginner level in Go. This stark contrast
between specialized AI systems and general-purpose LLMs has become a significant constraint
limiting LLMs' applications across broader scenarios.

---

[†]Corresponding Author.

39th Conference on Neural Information Processing Systems (NeurIPS 2025).

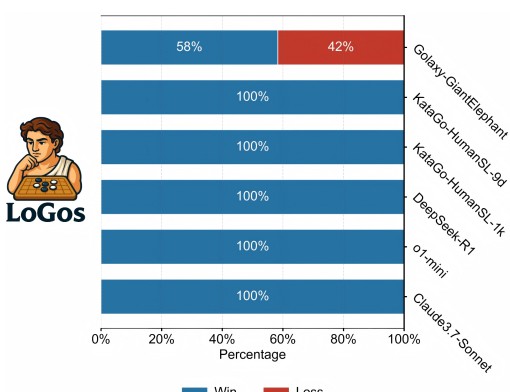

Figure 1: Win rates of **LoGos** against various models. For general LLMs, we select DeepSeek-R1, o1-mini, and Claude3.7-Sonnet for game-play. Additionally, we play LoGos against several specialized Go models. The KataGo-HumanSL model series [Wu, 2019] is designed to simulate human players at different skill levels. KataGo-HumanSL-1k emulates intermediate amateur players, while KataGo-HumanSL-9d mimics top amateur and professional players. Golaxy-GiantElephant is a well-known Go AI model that achieves performance comparable to mid-level professional players.

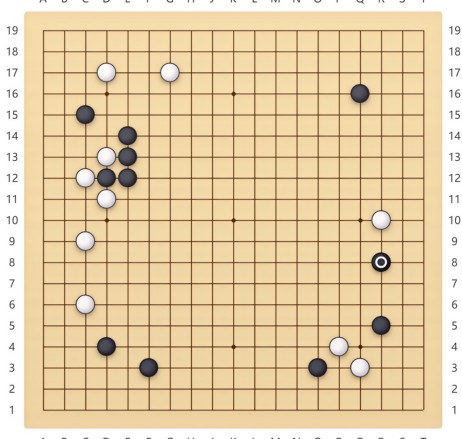

Figure 2: Given a game state, the model first analyzes the situation: "In the upper left corner: Black has formed a structure...", and then choose the valid next move **R8**.

Achieving expert-level performance in specialized domains with scarce pre-training corpora presents considerable challenges for general large language models. The scarcity of domain-specific data makes it virtually impossible for general LLMs to acquire Go strategies and master the specialized terminology through pretraining alone. Consequently, approaches based on direct distillation of general LLMs or pretraining on large-scale domain-specific natural language corpora [Liu et al., 2025, Wu et al., 2024] face limitations in the context of Go. Unlike tasks such as mathematics and coding, which benefit from abundant natural language corpora and human reasoning examples, specialized tasks like Go typically offer only structured domain-specific professional data that can be obtained at scale. Although some works have demonstrated that LLMs can complete professional tasks such as Chess through structured prediction [Feng et al., 2023, Schultz et al., 2024], a model that produces only structured outputs still falls short of the general reasoning model we aspire to develop.

If the necessary condition for achieving natural reasoning is to introduce a large amount of human reasoning data, then for a field like Go where such data is scarce, integrating professional capabilities into a general model will inevitably require extensive annotation efforts. This raises a critical question: Does it mean that incorporating specialized domain capabilities into general models is destined to remain an unresolved challenge?

To this end, we explore an alternative approach and discover a viable pathway forward. We assume that while general LLMs perform inadequately on unseen tasks due to their lack of specialized knowledge, they still demonstrate some reasoning generalization capabilities on novel tasks, attempting to understand, reason, and think based on task descriptions. Furthermore, while specialized domains like Go often have limited natural language corpora, they typically possess substantial amounts of structured domain knowledge. Therefore, we propose a novel training paradigm, which first injects domain expertise into general models through heuristically constructed expert-level synthetic data at scale, and then aligns this knowledge with the model's inherent reasoning capabilities.

Through heuristic rules, we construct a large-scale expert-level Go dataset, comprising a next-step prediction dataset based on structured domain-specific information and a Go commentary dataset.

Subsequently, we mix this specialized Go dataset with long chain-of-thought (CoT) reasoning data from domains including code and math for fine-tuning, achieving both the injection of domain expertise and the cold start of long CoT reasoning capabilities. Following this cold start phase, we employ Group Relative Policy Optimization (GRPO) [Shao et al., 2024b], encouraging the model to self-explore using long CoT reasoning forms for the next-step prediction task in Go.

As expected yet interestingly, through such reinforcement learning (RL) process, the fine-tuned model spontaneously self-explores and develops a stable strategy capable of natural reasoning and prediction based on correct understanding of board states. Our model achieves performance comparable to professional Go players, becoming the first general LLM to reach this level of proficiency. Further, the model maintains outstanding performance on general reasoning tasks such as math and code, becoming a strong generalized AI model that expert models such as AlphaGo cannot be. Our experiments conclusively demonstrate that general LLMs can acquire expert-level capabilities in a specific domain like Go, while maintaining general performance. We firmly believe this method can be applied to other domains and anticipate its broad application prospects. As alpha is the first letter in the Greek alphabet, AlphaGo heralded the beginning of a new era in artificial intelligence. In this work, we name our model **LoGos** (Language-Oriented Go System), which means "word" and "thought" in Greek, aspiring to bring the uniquely human capacity of reasoning back to the ancient game of Go. In summary, this paper makes the following contributions:

1. We introduce LoGos, a general LLM that achieves both expert-level proficiency in the domain of Go and outstanding performance in general reasoning tasks, demonstrating that LLMs can attain expert-level performance in specific domains while maintaining general reasoning capabilities.

2. We propose a training strategy that enables general models to achieve expert-level performance in specific domains using only structured domain-specific data which is scalable. We thoroughly validate the effectiveness of this strategy on Go.

3. We collect, construct, and release the first large-scale Go dataset specifically designed for LLM training, as well as an evaluation benchmark on Go designed for assessing LLMs.

## 2 Mixed Learning of Go

In this section, we detail our methodology for enabling general large language models to absorb expert-level ability in domains exemplified by the game of Go.

We begin by outlining our approach to modeling the next step prediction task within the Go domain. Subsequently, based on our collected and curated large-scale Go dataset, we construct a large-scale Go professional dataset covering two tasks: next move prediction and commentary. This dataset is consisted of a 10-million-scale next-step prediction dataset constructed using heuristic rules, alongside a 100K-scale commentary dataset.

Following this, we mix the Go professional dataset with general long-chain-of-thought (CoT) reasoning data from various domains such as code and mathematics. We fine-tune our base model on this mixed dataset to cold-start both Go-specific expertise and long-form CoT reasoning capabilities. Using this fine-tuned model as the starting point for reinforcement learning, we encourage self-exploration through carefully designed queries and rewards that promote long CoT reasoning on the next move prediction task. We design a segmented reward function to encourage the output of relatively optimal predictions in a long COT format. In this way, exemplified by the Go game, LLMs can self-explore expert-level reasoning chains.

### 2.1 Modeling the Game of Go

We first introduce our approach to modeling the next move prediction task in Go. As a highly complex board game, Go is played on a 19×19 grid, where black moves first and players alternate placing stones until the game ends. The complexity of board games can be quantified as $b^d$, where $b$ denotes the average number of

```
Given the Go game move list below:
1.X-Q16 2.O-D16 ... 141.X-J7
Please predict the next move.
```

Figure 3: Prompt demonstration for modeling the Go game as a next move prediction task.

legal moves per position and $d$ represents the typical game length. With parameters of $b \approx 250$ and $d \approx 150$, Go's complexity substantially exceeds that of other board games like chess ($b \approx 35$, $d \approx 80$), presenting a significantly more challenging task.

In this work, we formulate the entire sequence of moves in a Go game as the input sequence, and the model's task is to predict the next move. Specifically, for a Go position after $k$ moves, if we denote the preceding move sequence as $\{x_n\}_{n \leq k}$, then for a model $\pi_\theta$, the prediction task can be characterized as:

$$x_{k+1} = \pi_\theta(x_1, x_2, \ldots, x_k) \tag{1}$$

Each move position is a combination of a letter and a number coordinate. To distinguish between the two players, we use $X$ and $O$ to denote black and white stones, respectively. Therefore, given a game record sequence, the model is provided with a query as demonstrated in Fig. 3.

## 2.2 Expert-Level Go Dataset Collection

We introduce a large-scale professional Go dataset, comprising a Go commentary training dataset and a professional Go dataset constructed using heuristic rules for the next-step prediction task.

**Next Step Prediction Dataset** We collect a dataset containing over 5 million game records played by both top amateur and professional Go players. From these game records, we uniformly sample over 10 million game states and annotate them using the open-source Go engine KataGo [Wu, 2019]. As a Go engine that employs Monte Carlo Tree Search (MCTS), KataGo's output for a given game state includes the top 10 most probable next moves, subsequent variations for each move, and numerical analysis of the current move. We use KataGo to annotate these 10 million game states and design the following heuristic template to construct data for model training as seen in Fig. 4.

```
<Step1> (player identity determination) The
last move is 141.X-J7.  The next player is
white.
<Step2> (analysis of several potential next
moves) Try 142.O-C10, the subsequent possible
variation would be:  ...
<Step3> (summary) Considering all the above
analysis, the best move is 142.O-C10.
<Step4> (structured output)

   Next player:  White
   Next position:  C10
   Win rate:  52.4 %
```

Figure 4: Examples of the next step prediction dataset. The heuristic template consists of four parts: (i) confirming whether the next player is black or white; (ii) analyzing several possible next moves; (iii) summarizing and selecting the optimal next move; and (iv) structured output.

**Commentary Dataset** We collect and process 100K Go commentary cases from open resources, each containing an independent game state and the corresponding comment. We process the commentary data into a training dataset where, given a game state, the model's task is to provide logically sound commentary with correct terminology usage according to the current game state.

## 2.3 Mixed Cold Start

We mix the professional Go dataset with long CoT reasoning data from domains including mathematics and code, performing mixed fine-tuning on the base model to serve as the starting point for GRPO. After fine-tuning, LoGos demonstrates the capability to complete next-step prediction tasks and provide commentary based on the current game state. Simultaneously, the model maintains outstanding performance on other general reasoning tasks.

## 2.4 Self-Exploration with Reinforcement Learning

Based on our analysis of the fine-tuned initial model, we observe that it spontaneously generalizes reasoning capabilities acquired from long COT reasoning data to the Go prediction task. When presented with instructions resembling those in CoT reasoning contexts, the model naturally performs understanding, analysis, reasoning, and summarization for a given game state. However, at this stage, the initial model still cannot apply the professional knowledge acquired from the professional Go

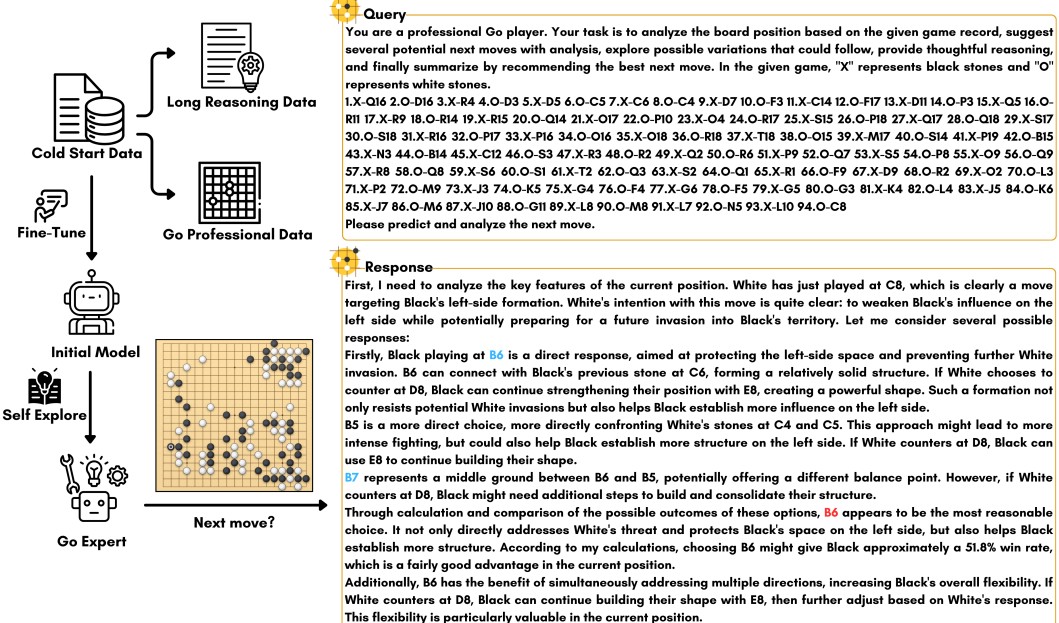

Figure 5: Our methodology for integrating Go professional capabilities with LLMs' long COT reasoning abilities. After mixed cold start and GRPO training, our model ultimately successfully transfers the reasoning capabilities acquired from long CoT data to Go tasks. For a given query, the model correctly performs analysis, thinking, reasoning, and summarization, ultimately selecting a reasonable next move.

dataset, resulting in inaccurate prediction results. To address this issue, we introduce a reinforcement learning phase to enhance the model's performance on the next-step prediction task. We uniformly sample from the KataGo annotated data described in Section 2.2, construct queries according to the game states, and utilize annotation information as metrics for reward calculation.

### 2.4.1  RL Optimization with GRPO

We utilize a widely used RL method, GRPO (Group Relative Policy Optimization), which is proposed in DeepSeek-Math. For a given problem-answer pair $(q, a)$, GRPO samples a group of independent responses $\{o_i\}_{i=1}^G$ from the old policy $\pi_{\theta_{old}}$. Each output is scored by a reward model or reward function, yielding $G$ rewards $r = \{r_1, r_2, \ldots, r_G\}$ correspondingly. GRPO optimizes the LLM by maximizing the following objective:

$$
\begin{aligned}
\mathcal{J}_{GRPO}(\theta) = \mathbb{E}[q \sim P(Q), \{o_i\}_{i=1}^G \sim \pi_{\theta_{old}}(O|q)] \\
\frac{1}{G} \sum_{i=1}^G \frac{1}{|o_i|} \sum_{t=1}^{|o_i|} \left\{ \min\left[ \frac{\pi_\theta^{i,t}}{\pi_{\theta_{old}}^{i,t}} \hat{A}_{i,t}, \text{clip}\left( \frac{\pi_\theta^{i,t}}{\pi_{\theta_{old}}^{i,t}}, 1-\epsilon, 1+\epsilon \right) \hat{A}_{i,t} \right] - \beta \mathbb{D}_{KL}\left[ \pi_\theta || \pi_{ref} \right] \right\}
\end{aligned}
\tag{2}
$$

where $A_i = \frac{r_i - mean(r_1, r_2, \ldots, r_G)}{std(r1, r2, \ldots, r_G)}$ is a group-relative advantage for the i-th response.

### 2.4.2  Reward Modeling

Specialized for the Go task, we design a reward function that rewards both high-matching moves and responses that predict the win rates accurately. We sample a subset from KataGo's annotated data, and for the top 10 options ranked by KataGo's win rate, we obtain and sort their win rates $\{w_i\}_{i=1}^{10}$, denoting the highest win rate as $w^*$. We stipulate that the model's output must include the predicted next move position along with a win rate estimation for that prediction. The win rates predicted by the model are denoted as $\{\hat{w}_i\}_{i=1}^{10}$. Based on this design, for a given model output $o_i$, the corresponding reward $r_i$ is calculated using a piecewise reward function:

$$r_i = \begin{cases} 1 - \alpha_1 \cdot \frac{\beta_1|(\hat{w}_i - w_i)|}{1 + \beta_1|(\hat{w}_i - w_i)|} & \text{if } rank(i) = 1 \\ c_1 - \alpha_1 \cdot \frac{\beta_1|(\hat{w}_i - w_i)|}{1 + \beta_1|(\hat{w}_i - w_i)|} - \alpha_2 \cdot \frac{\beta_2(w^* - w_i)}{1 + \beta_2(w^* - w_i)} & \text{if } rank(i) \in [2, 3] \\ c_2 - \alpha_1 \cdot \frac{\beta_1|(\hat{w}_i - w_i)|}{1 + \beta_1|(\hat{w}_i - w_i)|} - \alpha_2 \cdot \frac{\beta_2(w^* - w_i)}{1 + \beta_2(w^* - w_i)} & \text{if } rank(i) \in [4, 10] \\ c_3 - \alpha_1 - \alpha_2 & \text{if } rank(i) \notin [1, 10] \land \text{format correct} \\ 0 & \text{otherwise} \end{cases} \quad (3)$$

Here, $\alpha_1$, $\alpha_2$, $\beta_1$, $\beta_2$, $c_1$, $c_2$, $c_3$ are all adjustable parameters, satisfying $c_1 > c_2 > c_3 > \alpha_1 + \alpha_2$. We heuristically set $c_1 = 0.8$, $c_2 = 0.6$, $c_3 = 0.4$ to explicitly reward moves with higher matching degrees. We set $\alpha_1 = 0.1$, $\beta_1 = 10$ to reward responses with accurate win rate estimations for the predicted next move. Meanwhile, $\alpha_2$ is set to 0.2, with $\beta_2 = 10$, to reward relatively more matching predictions within the same ranking level.

With an external reward model such as KataGo, we collect structured expert-level knowledge, which is rather easy in expert-level reasoning tasks such as exploring scientific discoveries, data analysis, or controlling systems in the industrial process. By self-exploring the policies with RL training, the hard-to-generalize expert knowledge can be utilized in a general, powerful reasoning-style LLM.

## 3 Experiments

### 3.1 Experimental Setup

**Datasets** For the selection of long CoT reasoning data, we collect several distilled reasoning datasets covering a wide range of general tasks including code, mathematics, and general reasoning. Specifically, our collected datasets include Openthoughts-114K [Team, 2025], NuminaMath-QwQ-CoT-5M [Team et al., 2025], OpenCodeReasoning [Ahmad et al., 2025], Bespoke-Stratos-17k [Labs, 2025], and AM-DeepSeek-R1-Distilled-1.4M [Zhao et al., 2025].

**Benchmarks** We propose **KataGo-Bench-1K**, our original benchmark for measuring LLMs' Go capability. KataGo-Bench-1K is a test set of 1,000 samples from KataGo annotation data, with game states sampled across various player skill levels. For each position, a prediction is considered correct if the predicted move falls within the annotated candidate moves. Additionally, we measure the model's general performance on the following comprehensive benchmarks: **GPQA-Diamond** [Rein et al., 2024], **BBEH** [Kazemi et al., 2025], **KOR-Bench** [Ma et al., 2024a], **AIME**, **MATH** [Hendrycks et al., 2021], and **LiveCodeBench** [Jain et al., 2024]. These benchmarks collectively evaluate model performance across major reasoning domains, including commonsense reasoning, scientific knowledge, logical reasoning, mathematics, and coding abilities.

**Models** For foundation models, we conduct training based on the Qwen2.5 series [Yang et al., 2024], including Qwen2.5-7B-Base and Qwen2.5-32B-Base. For general performance comparison baselines, we select Qwen2.5-7B-Instruct and Qwen2.5-32B-Instruct, while also comparing with DeepSeek-R1-Distill-Qwen-7B and DeepSeek-R1-Distill-Qwen-32B [Guo et al., 2025] to thoroughly demonstrate our models' outstanding performance on general tasks. Furthermore, we evaluate several mainstream closed-source and open-source models for comparison, including DeepSeek-R1, OpenAI-o1-mini, and Claude3.7-Sonnet. We also compare Go professional capabilities with the KataGo-Human-SL model series [Wu, 2019], which is trained on human player game records. Specifically, this model series encompasses skill levels from 18k (kyu) to 9d (dan), with 27 distinct proficiency levels arranged sequentially from 18k to 1k, 1k to 1d, and 1d to 9d, where 18k represents beginner-level players and 9d represents top amateur and professional standards.

### 3.2 Main Results

Table 1 presents our primary experimental results. First, both our 7B and 32B model achieve Go-specific performance that significantly surpasses all existing general LLMs. On KataGo-Bench-1K, the strongest general model apart from LoGos is Claude3.7-Sonnet, which achieves a prediction accuracy of 34.3%. However, even this performance falls substantially below that of the lowest-ranked beginner-level Go specialized model, KataGo-HumanSL-18k (67.4%). In contrast, our

Table 1: Main results on Go professional benchmark and general benchmarks. We evaluate and compare the models' performance on KataGo-Bench-1K and several general reasoning benchmarks, with the evaluation results on general benchmarks obtained using Opencompass [Contributors, 2023]. The score on BBEH is calculated by averaging the results from each subset.

| Model | KataGo-Bench | General Benchmark | | | | | |
|---|---|---|---|---|---|---|---|
| | | GPQA Diamond | BBEH | KOR-Bench | AIME | MATH | LiveCodeBench |
| *Closed-source Models* | | | | | | | |
| DeepSeek-R1 | 17.6 | 69.7 | 44.5 | 78.3 | 86.7 | 97.6 | 83.8 |
| o1-mini | 27.3 | 61.1 | - | 70.16 | 56.7 | 95.0 | 75.0 |
| Claude3.7-Sonnet | 34.3 | 67.7 | 33.8 | 64.8 | 30.0 | 79.8 | 63.2 |
| *Open-source Models (7B)* | | | | | | | |
| Qwen2.5-7B-Base | 1.4 | 28.3 | 9.9 | 20.4 | 13.3 | 83.2 | 9.0 |
| Qwen2.5-7B-Instruct | 8.0 | 39.9 | 12.9 | 42.8 | 3.3 | 92.6 | 16.2 |
| DeepSeek-R1-Distill-Qwen-7B | 0.6 | **41.4** | 13.1 | 55.9 | 33.3 | 88.2 | 20.4 |
| **LoGos(7B)** | **88.1** | 37.9 | **22.1** | **65.7** | **40.0** | **93.2** | **23.4** |
| *Open-source Models (32B)* | | | | | | | |
| Qwen2.5-32B-Base | 1.5 | 35.9 | 15.6 | 42.8 | 10.0 | 90.7 | 13.8 |
| Qwen2.5-32B-Instruct | 6.8 | 46.0 | 18.0 | 58.1 | 20.0 | 95.2 | 29.9 |
| DeepSeek-R1-Distill-Qwen-32B | 4.7 | 56.1 | 27.4 | 70.0 | 46.7 | 94.5 | 36.5 |
| **LoGos(32B)** | **88.6** | **63.6** | **34.1** | **74.8** | **56.7** | **96.5** | **50.9** |
| *Go Professional Models* | | | | | | | |
| KataGo-HumanSL-18k | 67.4 | - | - | - | - | - | - |
| KataGo-HumanSL-1d | 79.7 | - | - | - | - | - | - |
| KataGo-HumanSL-5d | 85.5 | - | - | - | - | - | - |
| KataGo-HumanSL-9d | 87.8 | - | - | - | - | - | - |

Figure 6: Experiments on model self-exploration in the professional Go task. (a) Performance curve during reinforcement learning from our initial point in the main experiment; (b) Performance when applying reinforcement learning directly to instruction models without cold start; (c) Comparison of reinforcement learning performance when replacing heuristic rule construction with direct prediction during cold start.

models achieve nearly 2.6 times the accuracy of Claude3.7-Sonnet and even exceed the performance of KataGo-HumanSL-9d (88.6% and 87.8%), indicating that our models attain proficiency in Go comparable to professional players.

To verify that LoGos maintains strong general capabilities, we conduct comprehensive evaluations on various general benchmarks. The results demonstrate that LoGos not only excels in Go-specific tasks, but also maintains impressive performance across diverse general tasks, including mathematics, coding, and commonsense reasoning, achieving leading performance within similar model sizes. Compared to the Qwen2.5 series Instruct models, LoGos consistently outperform across all general tasks evaluated. Even when compared with DeepSeek-R1-Distill-Qwen-7B and DeepSeek-R1-Distill-Qwen-32B, our models achieve comparable performance across general tasks, surpassing the R1 distilled models on most of the benchmarks. These results conclusively demonstrate that our models have attained expert-level proficiency in Go while preserving outstanding general capabilities.

## 3.3 Experiments on Self-Exploration

### 3.3.1 Self-Exploration in Go

We analyze the performance growth of LLMs in Go through reinforcement learning. In Fig. 6(a), we plot the performance curves during GRPO training, using mixed fine-tuned Qwen2.5-7B-Base and

Qwen2.5-32B-Base as initial points. At the beginning of training, the models' initial performance ranges between 60% and 70%, comparable to beginner-level. As the training progresses, both of the models quickly discover a stable reasoning strategy through self-exploration, and their performance rapidly increases, surpassing the KataGo-HumanSL-1d model (representing an medium-level player). Subsequently, the models' performance continues to improve throughout the training process, eventually exceeding the KataGo-HumanSL-9d model, which simulates top human players and professional players, ultimately achieving a proficiency level comparable to professional Go players. The experimental results presented above demonstrate that after the cold start phase, LLMs are capable of exploring reasonable strategies for Go-related tasks through self-exploration.

### 3.3.2 RL without Cold Start

Given that models can self-explore in Go tasks, a natural question arises: can LLMs discover stable strategies through self-exploration without requiring a cold start? To investigate this question, we select the Qwen2.5-7B-Instruct and Qwen2.5-32B-Instruct, along with DeepSeek-R1-Distill-Qwen-7B and DeepSeek-R1-Distill-Qwen-32B as initial points, and directly apply GRPO training on the Go task. As seen in Fig. 6(b), our results confirm that although models can improve prediction accuracy throuth self-exploration, the upper limit is significantly lower without cold start, as the models ultimately fail to reach the beginner-level performance (67.36%). Notably, the self-exploration ceiling for DeepSeek's distilled models is lower than that of Qwen's instruction models, demonstrating that cold starting with long reasoning data alone does not directly result in effective generalization in domain-specific tasks like Go. Based on these observations, we conclude that general reasoning data alone is insufficient, and domain-specific expertise is necessary for effective performance in specialized tasks like Go.

### 3.3.3 Cold Start with Naive Fitting

While we have proved the significance of cold start, the necessity of heuristic rules when constructing expert-level dataset still needs verification. To investigate this, we try completely removing heuristic rules and use direct prediction as a substitution. For a given board state, the response directly predicts the position of the next move. As shown in Fig. 6(c), results on the 7B-size model show that when using naive fitting for cold start, the model fails to properly acquire Go expertise from the professional dataset, resulting in significantly lower capability ceiling (below 50%) compared with using heuristic rules (88%). These results confirm that in the cold start phase, a well-structured and relatively natural domain-specific expert-level dataset is crucial for helping LLMs raise their exploration ceiling during the self-exploration stage.

## 4 Discussion

### 4.1 Ablation Study on Reward Design

Since our reward function is heuristically designed, it requires further investigation. We conduct RL experiments with several alternative reward designs: (i) rewards given only to the top-1 move; (ii) rewards given only to the top-3 moves without distinguishing; and (iii) tiered processing of moves without intra-level rewards (i.e., setting $\alpha_1$ and $\alpha_2$ to zero in Equ. 3).

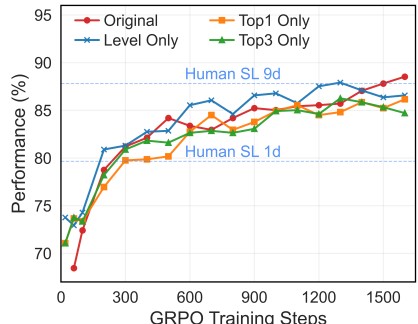

Figure 7: Ablation on reward design

As shown in Fig. 7, models perform effective self-exploration under different reward settings. However, under the top-1 and top-3 reward designs, the sparse reward distribution results in lower performance compared to the tiered reward and our original reward function. Comparing the original reward function with the tiered reward approach, no significant difference is observed, though our reward design ultimately achieved a higher performance ceiling, demonstrating the advantage of a smoother reward function.

Table 2: Human evaluation results on the generated responses, analyzing whether the predicted move is correct, and whether the corresponding explanation is correct, incorrect, or ambiguous.

| Move
Explanation | Correct | Incorrect |
|---|---|---|
| Correct | 121 | 4 |
| Ambiguous | 60 | 1 |
| Incorrect | 37 | 3 |

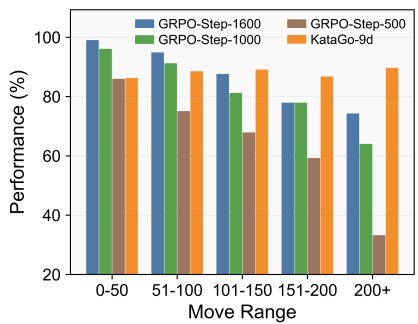

Figure 8: Analysis on move range

## 4.2 Case Study on Response Quality

We invite several top amateur Go players to evaluate the generated responses based on move prediction accuracy and explanation quality. As shown in Tab. 2, our model produces accurate move predictions in 96.5% of cases, with 55.6% of the explanations correctly given. However, there exist cases with incorrect explanations or ambiguous descriptions, which we will discuss briefly below.

Firstly, Go contains more than 600 terms, each corresponding to a specific local pattern on the board. Our model occasionally exhibits issues with incorrect usage of these specialized terms. Moreover, in certain scenarios, the correctness of the explanation becomes hard to assess. For example, during the opening phase, outputs such as "establishing strong positional influence" represent a generally applicable description that, while broadly valid, offers limited informational value.

## 4.3 Context Curse for LLM Performance in Go

### 4.3.1 Problem Description

As shown in Fig. 8, we observe that it becomes increasingly challenging for LLMs to comprehend the game state as the game lengthens. For a given game sequence $\{x_n\}$, any substitution between elements $x_i$ and $x_j$ could possibly create entirely different board positions. As the sequence length grows, the board structure becomes increasingly complex, requiring the model to capture more connections, which leads to decreased prediction accuracy. In contrast, models that directly represent the current board state, like KataGo, maintain performance regardless of game length.

### 4.3.2 Solution to context curse

To address this issue, we further explore encoding methods for the game records and design a strategy that incorporates 2-D board state information directly into the input query. Specifically, for a given move list, we provide the model with the resulting board state. This state is represented as a 19×19 2-D array, where the values 1, -1, and 0 denote a black stone, a white stone, and an empty intersection, respectively. Our Python implementation is based on an open-source Go board rendering repository https://github.com/SabakiHQ/go-board.

After modifying the query format and following the same experimental procedure (cold start + RL), we surprisingly find that the new approach significantly mitigates the model's "context curse" problem. Fig. 9 presents the model's performance under the new format across various input sequence lengths. It is evident that with this updated input,

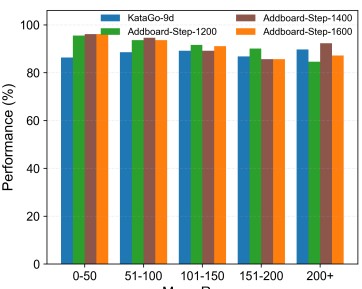

Figure 9: Analysis on move range (with 2D board rendering applied.)

the model achieves a markedly more accurate understanding of the board state, maintaining high prediction accuracy even for sequences exceeding 200 moves. Therefore, we conclude that this new method (2-D board state rendering) effectively resolves the context curse challenge.

### 4.4 Experiments on Data Mixing Ratio

#### 4.4.1 Impact of Data Mixture Ratios on RL Performance

We first investigate the impact of the data mixture ratio of Go dataset during the cold-start phase. While holding the training for the general reasoning dataset to 1 epoch, we compare the performance of models initialized with 1 versus 2 epochs of Go data. The models' performance is evaluated on the KataGo-Bench-1K benchmark throughout the subsequent RL stage.

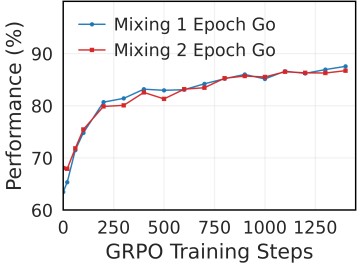

Figure 10: RL performance of different Go data mixing ratios.

As shown in Fig. 10, the model initialized with 2 epochs of Go data exhibits better performance at the initial point. However, as the RL process unfolds, this model does not demonstrate superior learning capabilities. In contrast, the model trained on a single epoch of Go data ultimately achieves a higher performance ceiling. Guided by this finding, and also considering the reduced computational requirements, we adopt the one-epoch setting for our cold-start experiment.

#### 4.4.2 Impact of Data Mixture Ratios on Generalization Performance

We then test the model's performance on general benchmarks after training for one epoch, mixing a fixed amount of reasoning data and varying quantities of Go data. As demonstrated in Tab. 3, incorporating a larger volume of Go data does not cause a significant degradation in the model's general reasoning capabilities.

Table 3: Performance on general benchmarks with varying amounts of Go data.

| Dataset | Without Go | 500K | 2M | 4M | 10M |
|---|---|---|---|---|---|
| MATH | 97.0 | 97.1 | 96.2 | **97.4** | 97.4 |
| AIME | **73.2** | 72.3 | 72.5 | 67.9 | 72.9 |
| OmniMath | 65.0 | 65.1 | **66.4** | 66.3 | 66.3 |
| GPQA diamond | **61.9** | 61.2 | 60.1 | 59.6 | 60.6 |

## 5 Related Works

**LLM as a Game Player**   Previous works have shown that LLMs demonstrate capabilities in various gameplay tasks. Through Chain-of-Thought (CoT) reasoning [Wei et al., 2022], LLM-based agents constructed through prompt design can complete gameplay tasks such as Pokémon battles, StarCraft II competitions, and Minecraft strategy simulations [Ma et al., 2024b, Hu et al., 2024, Madge and Poesio, 2024, Shao et al., 2024a, Gong et al., 2023, Chen et al., 2024]. However, tasks like Go are relatively more specialized, and general LLMs perform even worse than even beginners, rendering the approach of building agents for Go based on general LLMs . Some studies have explored the possibility of training LLMs to learn strategies for games such as chess and poker [Feng et al., 2023, Zhang et al., 2025, Huang et al., 2024]. Specifically for chess, further works have demonstrated that templates constructed with heuristic rules can help LLMs learn the trajectories and strategies generated by professional models in chess tasks [Wang et al., 2024, Schultz et al., 2024].

**Domain Expert AI**   In specialized domains, exemplified by AlphaGo and AlphaFold, AI systems have demonstrated expert-level capabilities [Silver et al., 2016, Jumper et al., 2021, Meng et al., 2024, Silver et al., 2017]. Building upon the powerful general capabilities exhibited by LLMs, several studies have explored methodologies for transforming LLMs into domain experts. A common paradigm involves synthesizing solutions to specialized problems into learnable data for language models based on heuristic rules. Leveraging tools such as the Lean language, works like AlphaGeometry [Trinh et al., 2024] and DeepSeek-Prover [Xin et al., 2024] have investigated approaches to equip LLMs with the capability of solving Olympic competition-level problems.

## 6 Conclusion

We present LoGos, a general large language models that achieves expert-level Go proficiency while maintaining outstanding general performance. We introduce a robust method to mix expert knowledge into general LLMs and demonstrate the potential of LLMs to acquire expert-level reasoning abilities in specific domains. We believe that in the future, such a mixing-expert-knowledge method can lead to a wider range of LLM applications in the AI industry.

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

# A  Limitations

First, we will discuss the limitations of this work. Although our method effectively and robustly integrates domain-specific expert-level knowledge into general LLMs, this approach still has several necessary preconditions for application. Taking Go as an example, our method requires a scalable source of professional knowledge, such as KataGo. Further, we need to design heuristic rules based on professional knowledge and construct cold start synthetic data that LLMs can learn from and understand, thereby training an initial point for self-exploration. In other specialized domains, acquiring such scalable professional knowledge might necessitate the introduction of domain-specific models/databases, while the design of heuristic rules should also consider the specific task requirements. Therefore, this method of mixing-expert-knowledge inevitably requires deep cooperation with researchers in specialized fields to enable broader applications.

Furthermore, despite the remarkable effectiveness of self-exploration, its training speed is relatively slower. Consequently, compared with traditional AI models in Go, LoGos is trained on significantly fewer game states during the RL phase. We therefore believe that the autoregressive architecture of LLMs might actually constrain the training efficiency when applied to tasks requiring real-time reasoning, such as Go.

# B  Experimental Details

## B.1  Experimental Settings and Compute Resources

In the mixed cold start phase, we SFT the base models with a maximum sequence length of 32,768 tokens. We employed a cosine annealing learning rate scheduler with rates ranging from 4e-5 to 4e-6. For training the 7B model, we utilized 32 A800 (80GB) GPUs, while the 32B model required 64 A800 GPUs.

In GRPO training stage, our implementation is primarily based on modifications to the VerL framework [Sheng et al., 2024]. Regarding specific parameter settings, we configure the training batch size to 64, with 16 roll-outs per data point and a maximum response length of 8,192 tokens. Due to the significant distribution gap between Go task responses and the reference model's pretraining data, we set the KL coefficient (kl_coef) to 5e-4. The 7B model training utilizes 32 A800 GPUs, while the 32B model reinforcement learning phase requires 64 GPUs.

## B.2  Evaluation Details

For general benchmarks, all evaluations are conducted using the OpenCompass framework. Specifically, the evaluation results for AIME and MATH datasets employ Qwen2.5-72B-Instruct as the judge model. The performance metrics reported on the BBEH benchmark represent the average performance across all the subsets.

# C  Further Discussions

## C.1  Analysis on KataGo-Bench-1K

Since most of our experimental results on Go are evaluated based on the performance on the constructed **KataGo-Bench-1K** benchmark, it is essential to discuss the fairness of KataGo-Bench-1K. We select several models with different proficiency levels to play against each other, and calculate the corresponding ELO ratings based on win-loss outcomes, thereby intuitively representing model capabilities through competitive relationships. The ELO rating is designed as follows:

$$\text{ELO}(A)_{\text{new}} = \text{ELO}(A)_{\text{old}} + K \cdot (S_A - E_A) \qquad (4)$$

Where:

- $\text{ELO}(A)_{\text{new}}$ is the updated Elo rating of player A after the match
- $\text{ELO}(A)_{\text{old}}$ is the previous Elo rating of player A

- $K$ is the weight coefficient that determines how much a single game will impact the rating (set to 32)
- $S_A$ is the actual score of player A in the match (1 for a win, 0.5 for a draw, 0 for a loss)
- $E_A$ is the expected score of player A, calculated as:

$$E_A = \frac{1}{1 + 10^{(\text{ELO}(B) - \text{ELO}(A))/400}} \tag{5}$$

Where $\text{ELO}(B)$ is the Elo rating of opponent B. The constant 400 determines how much rating difference corresponds to a specific winning probability.

Based on this ELO rating system, we selected several models with varying performance levels and conducted a tournament to obtain the corresponding ELO ratings for each model. As shown in Figure 11, our experimental results demonstrate that the correlation coefficient between the models' ELO ratings and their performance on KataGo-Bench-1K is $r = 0.92$, indicating an extremely high correlation between these two metrics. Therefore, we can conclude that KataGo-Bench-1K is a fair benchmark that accurately reflects the actual competitive capabilities of the models.

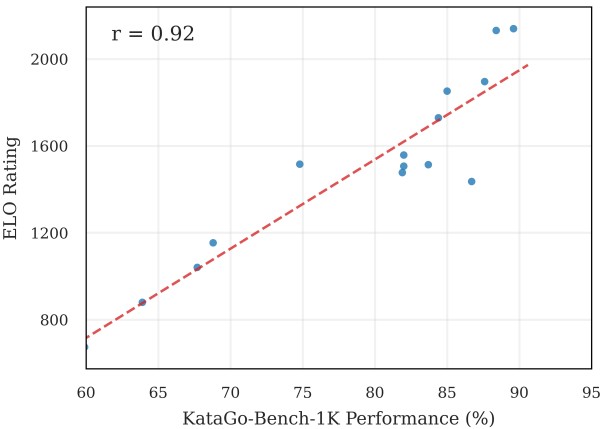

Figure 11: Correlation between ELO rating and KataGo-Bench-1K performance.

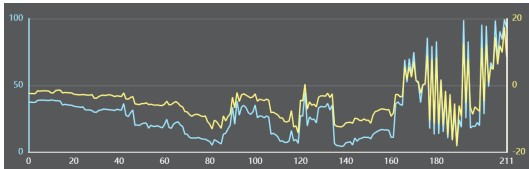

Figure 12: Winrate curve when context curse occurs

## C.2 Context Curse: A Case study

In Sec. 4.3, we introduce the phenomenon of declining LLM performance as game state length increases. Here, we will study on this phenomenon and its underlying causes through several games between LoGos and Golaxy-GiantElephant. Fig. 12 illustrates the win rate and score progression during gameplay. The blue line represents Golaxy-GiantElephant's win rate, while the yellow line indicates Golaxy-GiantElephant's lead in points. As observed, LoGos maintained a significant advantage during the first 160 moves. However, within the subsequent 40 moves, following several mistakes by LoGos, the game situation completely reversed, ultimately resulting in LoGos's defeat.

Examining one specific mistake, the board position is shown in Fig. 13. At this juncture, the green positions represent the top-1 choice annotated by KataGo, whereas the red or brown positions indicates LoGos's actual choice, which are incorrect. Black's previous move was at position H8.

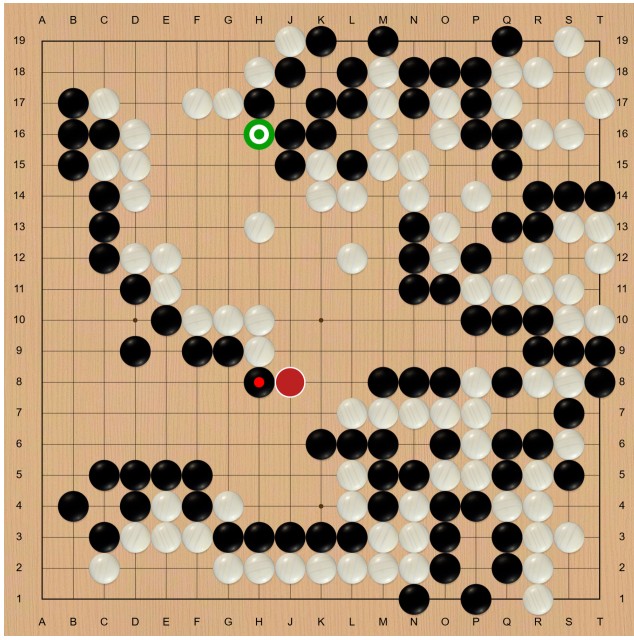

Figure 13: Game state when error occurs

LoGos's prediction included the reasoning: "We need to respond to Black's challenge at position H8." In Go, responding to an opponent's attack in the local area is a common strategy, making this pattern easily learnable from the training data. However, the superior move at H16 requires comprehensive value judgment. While such a decision is not particularly difficult for top amateur human players, it presents a significant challenge for LLMs due to their reliance on serialized input, making global pattern recognition more difficult. This obstacle is what we name as the "context curse" faced by LLMs in Go-related tasks.

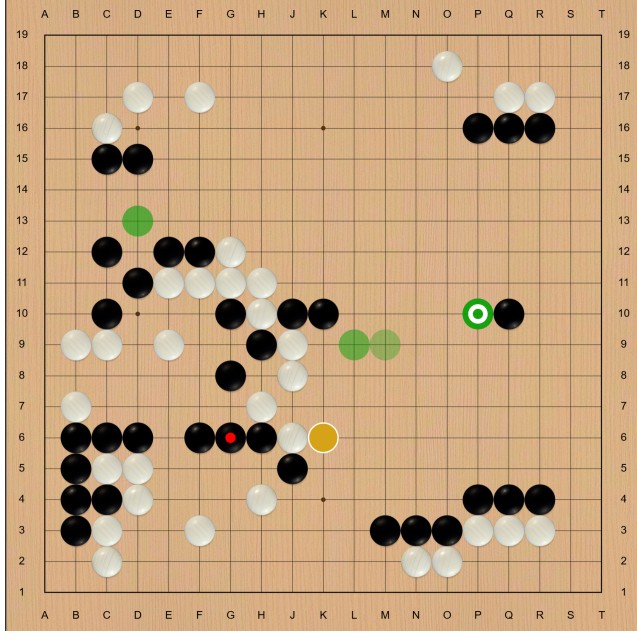

Figure 14: Another game state, LoGos chose to respond in the local area again

## C.3 Response Quality: More Case Studies

**Instructions Given to Volunteers** We invite more than twenty amateur Go enthusiasts to provide manual evaluations and ratings based on game states and model-generated responses. Our evaluation process is completely anonymous and randomly distributed, and the instructions do not have any privacy risks. The instructions are designed as follows:

```
Given the game state on the board and the response generated by our LLM, please evaluate the
accuracy of the model-generated response.  You need to consider both the correctness of the move
itself and the accuracy of the corresponding explanation.  The move should be categorized as
correct/incorrect, while the explanation should be classified into three categories:  correct,
incorrect, or ambiguous.  If you find it difficult to determine the accuracy of the current
response's explanation, please select the ambiguous category and provide your reasoning.
```

Figure 15: Instructions given to volunteers

**Case Studies on Successful Examples** In this part, we present several examples where the model predictions are accurate. During our experiments and annotation process, the model outputs were in Chinese. Here, we provide the translated version to facilitate understanding.

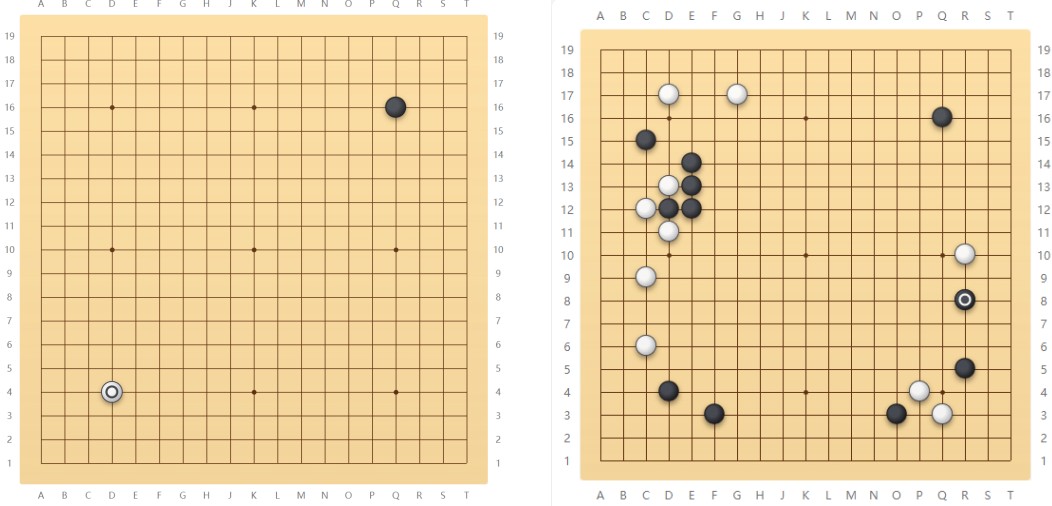

Figure 16: Case 1                    Figure 17: Case 2

First, Figure 16 illustrates a correct prediction made by the model at the beginning of the game. For this particular game state, the generated response is demonstrated in Fig. 18.

In the generated output, LoGos correctly predicts the next move position (D4) while accurately employing proper terminology: identifying D4 as a star point on the board. Furthermore, LoGos successfully conduct accurate variation analysis and result summarization, selecting D4 as the correct next move from among several possible variations. It also appropriately applies the Go principle of "enter slowly" to support its reasoning for the next move selection.

Figure 17 demonstrates another accurate prediction during the opening phase. This game state is sampled from a professional player's match. In this position, the model correctly identifies R8 as the optimal next move. The corresponding response is shown in Fig. 19. Within this response, the model again conducted comprehensive reasoning and variation analysis, selecting the correct next move through a natural and vivid presentation.

Next, we present several mid-game states and their corresponding responses. In a typical Go match, the period from approximately move 75 to move 200 is characterized by more intense confrontations and battles between the opponents.

As shown in Figures 20 and 21, these examples illustrate mid-game positions. In such situations, the model receives input sequences ranging from 75 to 150 moves in length, requiring it to model complex

Aha, Black's first move is Q16 at the star point, which is a very solid opening. As
White, we have multiple response options. Let's analyze them carefully.
In modern Go, facing a star point opening, White has three classic standard responses:
occupying the diagonal star point, playing the small point on the same side, or directly
attaching at the corner. Let's examine the specific variations:
1. White occupies the diagonal star point at D4 This is the most traditional response,
similar to the "diagonal stepping" concept in ancient Chinese military strategy. If we
choose this variation: 2.O-D4 3.X-Q3 4.O-D17 5.X-C15 6.O-E16 7.X-R6 8.O-C12 According
to AI analysis, White's win rate in this variation is 52.3%, which is quite balanced.
This approach maintains global flexibility, allowing for pivoting toward different
directions as needed.
2. White occupies the lower-left at D3 This choice is also reasonable, reflecting the
Go proverb "corners first, then sides." The variation might proceed as: 2.O-D3 3.X-Q3
4.O-D17 5.X-C5 6.O-C14 7.X-R6 8.O-C4 The win rate is similarly 52.3%, with this approach
emphasizing development toward the lower-left direction.
3. White chooses to attach at D16 This represents a more direct, aggressive approach:
2.O-D16 3.X-Q3 4.O-D3 5.X-C5 6.O-C4 7.X-D5 8.O-F3 9.X-C17 Although the win rate remains
at 52.3%, this approach is more aggressive and might complicate the game prematurely.
After careful consideration, I believe D4 is the optimal choice for three reasons:
1) It adheres to the fundamental Go principle of "corners first, then sides"; 2) It
maintains global balance; 3) It provides more possibilities for subsequent development.
Moreover, since Black has only played one move, there's no need to define the game
pattern too early. Choosing D4 aligns with the Go maxim "enter slowly," encouraging
patience while waiting for better opportunities.

---

Next player: White
Next position: D4
Win rate: 52.3%

Figure 18: Response demonstration for case 1

Hmm... this position is quite interesting. White's 20th move, the small knight's move
at P4, shows considerable insight-it both restricts Black's development around Q3 and
lays groundwork for future development in the lower right corner.
(Suddenly sitting upright) However, Black is not to be underestimated! Let's analyze
several key points carefully:
1. The contest for the lower right corner: R8 is particularly tempting. (Gesturing
over the board) If Black can establish a foothold here, it not only creates resonance
with R5 but also opens a gap in White's small knight's formation. I anticipate White
might attempt an invasion at O4, but Black can respond with N4, maintaining initiative
in the center.
2. Potential opportunity in the upper left corner: Q9 is also worth considering.
(Slightly shaking head) While it could construct an interesting shape, it's perhaps
too passive and doesn't exert sufficient direct pressure on White. The 46.6% win rate
indeed indicates this isn't the optimal choice.
3. Other possibilities: (Quickly surveying the board) Other points are either too
conservative or... (suddenly laughing) frankly speaking, they hardly qualify as proper
moves! Especially at this stage, Black needs an assertive approach.
(Rolling a black stone between fingers) After careful analysis, the R8 variation
proves most advantageous: 21.X-R8 22.O-O4 23.X-N4 24.O-N5 25.X-M4 (Striking the board
decisively) This allows Black to establish a solid bridgehead in the lower right!
Though the 47.4% win rate doesn't represent an overwhelming advantage, the game is still
long, and this choice at least enables Black to maintain initiative.
(Addressing an imaginary opponent) "White-san, your P4 move was indeed elegant, but my
R8 is equally commendable. Let's see who has the last laugh in the lower right corner!"

---

Next player: Black
Next position: R8
Win rate: 47.4%

Figure 19: Response demonstration for case 2

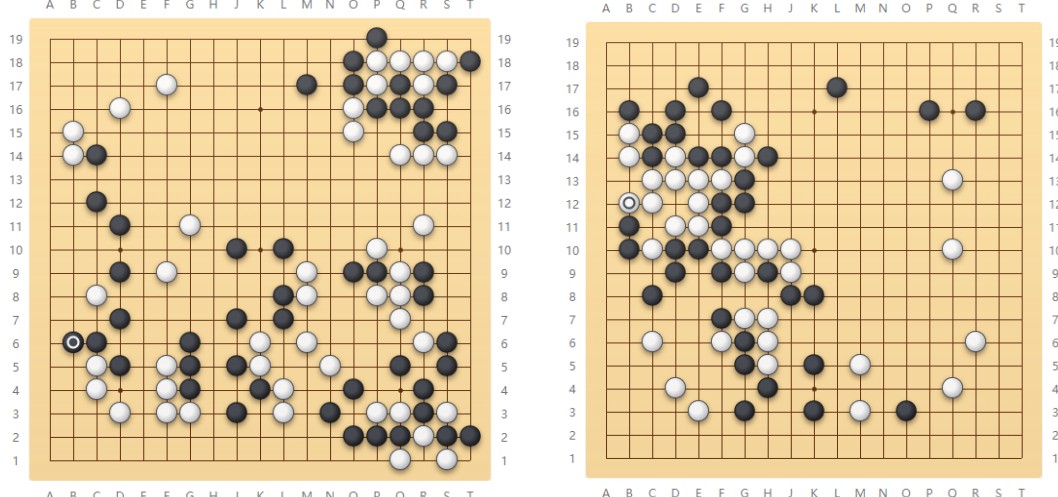

Figure 20: Case 3

Figure 21: Case 4

```
Hmm... Black's 69th move at B11 is quite sharp. This move has me somewhat perplexed,
as it simultaneously threatens my formation on the left and my development space in the
center-truly a "good move."
(Sitting upright, carefully examining the board) Let me analyze the current position.
The battle on the left side is critical; Black's B11 move clearly aims to gain
initiative in this area. As White, I need to respond cautiously.
(Taking up a fan and gently waving it) I've considered the following response options:
Option One: White B12 This move is direct and effective, embodying the Go principle of
"attack while defending oneself." B12 can resonate with the previous B14, constructing
a stable structure. Let's project the subsequent variations: 70.O-B12 71.X-E6 (Black
might counterattack here, attempting to establish influence in the center) 72.O-F5 (A
solid response, simultaneously expanding toward the center) 73.X-E5 74.O-F4 75.X-G4
76.O-E7 77.X-F8 (Nodding slightly) In this variation, White's win rate is 58.9%, making
it a favorable choice.
Option Two: White C11 (Frowning in contemplation) This move also appears promising,
but may lack directness. While C11 offers flexibility, it potentially leaves Black
with more opportunities: 70.O-C11 71.X-C9 72.O-B12 73.X-E6 74.O-F5 75.X-E5 76.O-F4
77.X-G4 (Shaking head gently) In this variation, White's win rate is 54.8%, slightly
less advantageous.
(Setting down the fan, crossing hands) Comparatively, B12 clearly prevails. It not
only directly addresses Black's threat but also accommodates development in multiple
directions, creating excellent synergy with B14. Should Black attempt complications
here, we have sufficient countermeasures.
(Slight smile) While the opponent's B11 move did cause some perplexity, our B12 response
should exert some pressure in return. This move adheres to the principle of "enter
slowly when approaching enemy territory" while maintaining our advantage.

   Next player: White
   Next position: B12
   Win rate: 58.9%
```

Figure 22: Response demonstration for case 3

positional relationships. During this phase, even a minor mistake at any moment can cause dramatic fluctuations in win probability. We present two cases where the model made correct predictions. Similar to Case 2, these game states are sampled from professional players' matches. According to human evaluation, these game states represent particularly critical phases of play. Figure 22 displays the model-generated response for the third game state demo. Notably, the model correctly addresses the challenge appearing on the left side, maintain its local structure, and appropriately applies various Go terminology and principles. In the demonstrated game state, the move chosen by LoGos (B12) is the only correct solution. Additionally, the model demonstrated the ability to generate colloquial and natural responses.

**Case Studies on Misuse of Go Terms**   Next, we examine cases that illustrate the two issues mentioned in Section 4.2: the model's incorrect usage of specific terminology in particular positions and the difficulty in evaluating the correctness of model-generated responses. First, Figure 23 shows a position from the opening stage. Here, the model correctly selects D17 as the predicted next move. However, the model erroneously describes D17 as a "star point," which actually refers to the position at D16 one intersection away. While this error does not affect the accuracy of the model's final prediction, it objectively reflects discrepancies in the model's understanding of specific terminology.

Similarly, in Figure 24, the model characterizes the relationship between the predicted next move F17 and the existing stone at D17 as a "knight's move," when the correct term should be "one-space jump"—another instance of terminology misuse. Overall, the model's terminology errors appear predominantly during the opening stage, likely because the opening phase involves a richer variety of shapes and associated terminology.

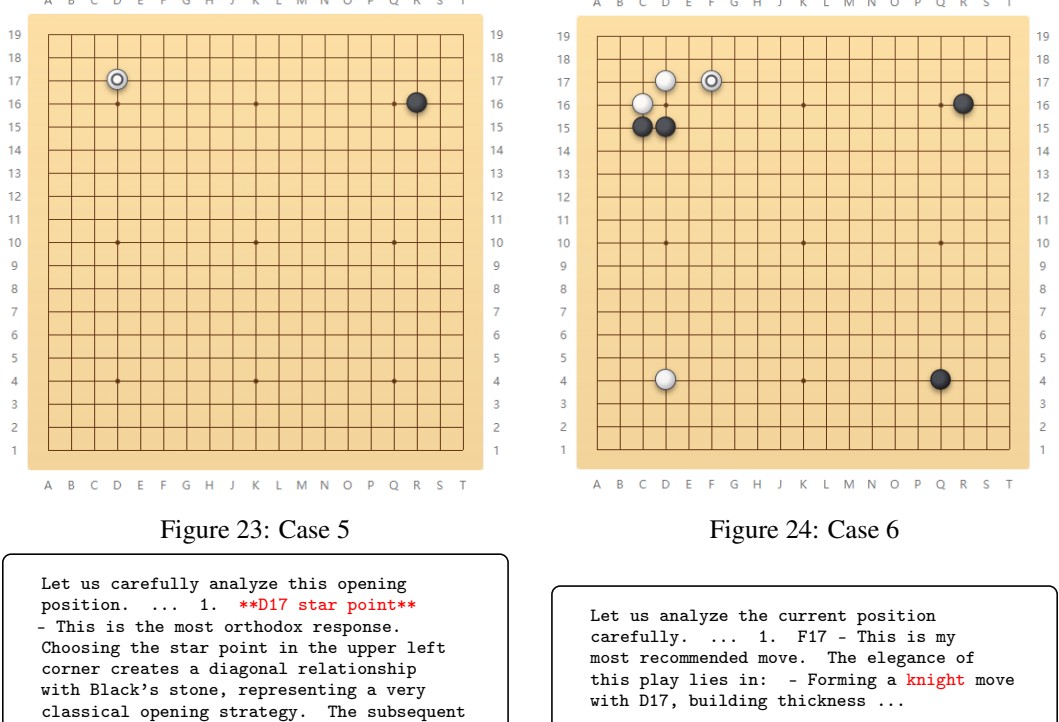

Figure 23: Case 5

Figure 24: Case 6

```
Let us carefully analyze this opening
position.  ... 1.  **D17 star point**
- This is the most orthodox response.
Choosing the star point in the upper left
corner creates a diagonal relationship
with Black's stone, representing a very
classical opening strategy.  The subsequent
variation might develop as follows:  ...
```

```
Let us analyze the current position
carefully.  ... 1.  F17 - This is my
most recommended move.  The elegance of
this play lies in:  - Forming a knight move
with D17, building thickness ...
```

Figure 26: Response demonstration for case 6

Figure 25: Response demonstration for case 5

**Case Studies on Ambiguous Explanations**   In Section 4.2, we also discuss instances where LoGos provides responses whose correctness is difficult to assess accurately. Below, we present a corresponding example in Fig. 27. In this position, the model's prediction at O4 would typically be described as cutting Black's connection in the lower right and punishing Black's overplay in this area. However, the model characterizes the effect of this move as "not only neutralizing Black's offensive, but also establishing a solid base for our side in the lower right." While this description is far from incorrect, from a response quality perspective, it provides relatively limited effective information and does not fully align with human-expected intent. Therefore, responses of this quality are considered ambiguous.

We consider the occurrence of this phenomenon to be quite reasonable. During the GRPO training phase, we do not incorporate any reward related to terminology usage for the model. Indeed, adding such a reward is not a straightforward task, as currently no language model possesses the capability to correctly understand Go board terminology, which implies that training a corresponding reward model would be an entirely new challenge. Nevertheless, it is surprising that despite not receiving any terminology-related rewards during training, LoGos still maintains a relatively high descriptive

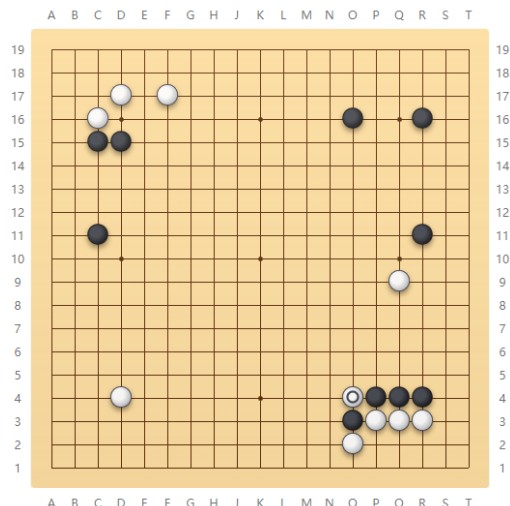

Figure 27: Case 7

Let us analyze the position carefully:
...  After careful deliberation, O4 is
undoubtedly the optimal choice.  It not
only neutralizes Black's offensive, but
also establishes a solid base for our
side in the lower right.  (Sitting down
again, confidently picking up a stone) It's
time to show Black what true defensive
counterattack looks like!

Figure 28: Response demonstration for Case 7

accuracy and successfully employs various specific terminologies acquired from the commentary dataset.

```
<Step1> (player identity determination) The last move is 141.X-J7.  The next
player is white.
<Step2> (analysis of several potential next moves) Try 142.O-C10, the
subsequent possible variation would be:  142.O-C10 143.X-C9 144.O-D9
145.X-D10 146.O-E9 147.X-E10 148.O-F9 149.X-F10 150.O-G9 In this variation,
white's win rate is 52.4%.
Try 142.O-D10, the subsequent possible variation would be:  142.O-D10
143.X-C10 144.O-C9 145.X-D9 146.O-E9 147.X-E10 148.O-F9 149.X-F10 150.O-G9 In
this variation, white's win rate is 51.8%.
Try 142.O-C9, the subsequent possible variation would be:  142.O-C9 143.X-C10
144.O-D10 145.X-D9 146.O-E9 147.X-E10 148.O-F9 149.X-F10 150.O-G9 In this
variation, white's win rate is 51.5%.
<Step3> (summary) Considering all the above analysis, the best move is
142.O-C10.  This move leads to a win rate of 52.4%, which is higher than the
alternatives.  The move at C10 effectively controls the left side of the
board while maintaining a balanced position overall.
<Step4> (structured output)

    Next player:  White
    Next position:  C10
    Win rate:  52.4 %
```

Figure 29: Examples of the next step prediction dataset. The heuristic template consists of four parts:
(i) confirming whether the next player is black or white; (ii) analyzing several possible next moves;
(iii) summarizing and selecting the optimal next move; and (iv) structured output.

# D   More Demonstrations of Prompts and Datasets

## D.1   Next Step Prediction Dataset

In Fig. 29, we present a complete response constructed based on heuristic rules from the Next Step
Prediction Dataset.

## D.2   Commentary Dataset

In Fig. 30, we show a case from the Go Commentary Dataset.  The comments are sourced from
professional and top amateur Go players' game analyses. We collect and process the data to establish

```
Move 76:  Next, Black could well jump to the star point at the bottom center.
In any case, due to the unfavorable position, Black is now seeking to employ
extraordinary measures.
```

Figure 30: Example of the Go commentary dataset

one-to-one correspondences between game states and comments, and apply rule-based filtering to remove low-relevance comments (such as greetings at the beginning of games or announcements of winners at the end). Through this process, we construct a Go Commentary Dataset comprising 100K entries.

## D.3  System Prompt in GRPO

During the GRPO training phase, we employed a system prompt entirely different from that used in the cold start phase, aimed at encouraging the model to think and predict through a reasoning paradigm. Our system prompt is shown in Figure 31.

```
You are a professional Go player.  Your task is to analyze the given game
record, assess the position, select several possible next moves for analysis,
project the subsequent variations, conduct reasonable analysis and reflection,
and finally summarize and select the optimal next move.  In the provided game,
"X" represents Black stones, and "O" represents White stones.  The board
size is 19x19, with each move coordinate expressed as a letter followed by
a number.  Letters range from A-T (skipping I), corresponding to positions
from left to right on the board.  Numbers range from 1-19, corresponding to
positions from bottom to top.
You need to first conduct a reasonable analysis and reflection on the current
position, make logical predictions, projections, and analyses of subsequent
moves, and finally summarize your reasoning to select the most appropriate
next move.  Please provide rigorous and detailed analytical reasoning, with
timely summaries.  Your output format should be:
<reasoning>
Your thought process.
</reasoning>
<answer>
\boxed{Next player: Black/White}
\boxed{Next position: move location}
\boxed{Win rate: percentage}
</answer>
```

Figure 31: Example of the system prompt used in GRPO

## E   2-D Board Rendering Demo

In this section, we briefly introduce the method we use to render a given move list to the 2-D board state.

For instance, for a move list:

```
1.X-D16 2.O-D4 3.X-Q4 4.O-Q16 5.X-O17 6.O-R14 7.X-C3 8.O-D3 9.X-C4 10.O-D5
11.X-B6 12.O-R6
```

Figure 32: Example of original move list.

The rendered 2-D board state is shown in Fig. 33. Where 1 represents black, -1 represents white, and 0 represents an empty position.

## F   More descriptions of Go

Here are some simple explanations for better understanding of Go.

```
[[0, 0, 0, 0, 0, 0, 0, 0, 0, 0, 0, 0, 0, 0, 0, 0, 0, 0, 0], [0, 0, 0, 0, 0, 0,
0, 0, 0, 0, 0, 0, 0, 0, 0, 0, 0, 0, 0], [0, 0, 0, 0, 0, 0, 0, 0, 0, 0, 0, 0, 0,
0, 1, 0, 0, 0, 0, 0], [0, 0, 0, 1, 0, 0, 0, 0, 0, 0, 0, 0, 0, 0, 0, 0, -1, 0, 0,
0], [0, 0, 0, 0, 0, 0, 0, 0, 0, 0, 0, 0, 0, 0, 0, 0, 0, 0, 0], [0, 0, 0, 0, 0, 0,
0, 0, 0, 0, 0, 0, 0, 0, 0, 0, -1, 0, 0], [0, 0, 0, 0, 0, 0, 0, 0, 0, 0, 0, 0,
0, 0, 0, 0, 0, 0, 0], [0, 0, 0, 0, 0, 0, 0, 0, 0, 0, 0, 0, 0, 0, 0, 0, 0, 0, 0,
0, 0], [0, 0, 0, 0, 0, 0, 0, 0, 0, 0, 0, 0, 0, 0, 0, 0, 0, 0, 0], [0, 0, 0, 0,
0, 0, 0, 0, 0, 0, 0, 0, 0, 0, 0, 0, 0, 0, 0], [0, 0, 0, 0, 0, 0, 0, 0, 0, 0, 0,
0, 0, 0, 0, 0, 0, 0, 0], [0, 0, 0, 0, 0, 0, 0, 0, 0, 0, 0, 0, 0, 0, 0, 0, 0,
0, 0, 0], [0, 0, 0, 0, 0, 0, 0, 0, 0, 0, 0, 0, 0, 0, 0, 0, 0, 0, 0], [0, 1, 0,
0, 0, 0, 0, 0, 0, 0, 0, 0, 0, 0, 0, -1, 0, 0], [0, 0, 0, -1, 0, 0, 0, 0, 0,
0, 0, 0, 0, 0, 0, 0, 0, 0, 0], [0, 0, 0, 1, -1, 0, 0, 0, 0, 0, 0, 0, 0, 0, 0, 0,
1, 0, 0, 0], [0, 0, 1, -1, 0, 0, 0, 0, 0, 0, 0, 0, 0, 0, 0, 0, 0, 0, 0], [0,
0, 0, 0, 0, 0, 0, 0, 0, 0, 0, 0, 0, 0, 0, 0, 0, 0], [0, 0, 0, 0, 0, 0, 0, 0,
0, 0, 0, 0, 0, 0, 0, 0, 0, 0, 0]]
```

Figure 33: Rendered 2-D board state.

- Go is a two-player game where one player uses black stones and moves first, while the other player uses white stones. The two players take turns placing stones.

- In Go, each stone must be placed on an intersection point. Each stone survives if and only if it has at least one "liberty" (breathing space). When a stone is surrounded by the opponent's stones on all adjacent sides, it is considered to have zero liberties and is "captured". Typically, a game ends when all stones from both sides are alive, and further moves cannot guarantee survival. At this point, both players agree to end the game and count the total number of their surviving stones plus the enclosed intersection points. The player with the higher total wins. For experienced players, the game can end simply by mutual agreement that all moves are complete and there is no more question about the life and death status of stones on the board, without needing to continue placing stones.

- "Capture" is a professional Go term that refers to the action of placing a stone that removes all liberties from the opponent's stones, thereby clearing the captured stones from the board. This is one of the most fundamental rules of Go.

# G  Broader Impacts

In this paper, we do not utilize any non-public data. All our game records and commentary data are obtained from open resources on the internet, containing only move information, comments, and necessary win-loss relationships, thus posing no privacy risks or negative societal impacts.

From a positive perspective, we propose a strong method for integrating domain-specific expert-level knowledge with general-purpose LLMs, providing a novel approach for a wider range of LLM applications in the AI industry.

