# OpenReview forum: "Mixing Expert Knowledge: Bring Human Thoughts Back To the Game of Go"
_NeurIPS.cc/2025/Conference — NeurIPS 2025 poster_

### Official Review · Reviewer_Dp9f · 2025-06-30

**Clarity:** 3
**Significance:** 2
**Originality:** 3
**Rating:** 3
**Confidence:** 2

**Summary:**

This paper presents LoGos, a large language model trained to achieve professional-level Go performance while maintaining general reasoning capabilities. The authors use a two-stage approach: (1) mixed fine-tuning with structured Go expertise data and general Chain-of-Thought reasoning data, followed by (2) reinforcement learning using GRPO. The model reportedly achieves 88.1% and 88.6%  accuracy on their KataGo-Bench-1K benchmark and wins against various Go AI models.

**Questions:**

As a Go game player, I genuinely like the topic of this paper.

I have one major question: Can you provide game records and independent evaluation against established Go engines at known strength levels? The reason I'm asking is that I tried to quickly reproduce the strategies of using text encoding with commercial LLMs (Claude, OpenAI, etc.), yet I do not really observe consistent responses from LLMs. In particular, when I provide the sequential text encoding steps of Go games to LLMs, I do not see reasonable plays from the responses. The model more likely just hallucinates and generates a random response. I don't know if this is because of the issues of not fine-tuned on Go-related materisl, but I would like to see more evidence that the approaches proposed by the authors work. Therefore, it would be great if the authors could open-source the dataset, or at least provide some samples.

1. Core Justification: Why use text encoding for Go when spatial representations are more natural? What advantages does this approach offer over existing methods?
2. Understanding vs. Pattern Matching: How do you distinguish between the model learning Go principles versus memorizing patterns from training data?
3. Conversation Logs: Can you share actual conversation histories showing the model's reasoning process during gameplay?
Spatial Reasoning: How does the model handle complex spatial relationships (life/death, territorial evaluation) through text sequences?
4. Generalization: How does performance vary across different board sizes or rule variants not seen in training?

**Ethical Concerns:**

["NO or VERY MINOR ethics concerns only"]

**Final Justification:**

I decide to keep my scoring as discussed in the comments previously. In particular, my questions and concerns on reproducibility is not addressed.

**Limitations:**

- The text encoding approach may fundamentally limit spatial reasoning capabilities
- Training methodology may overfit to evaluation metrics rather than genuine understanding
- Limited analysis of failure modes and edge cases
- Dependence on KataGo annotations may constrain learning to that engine's biases
- Computational efficiency compared to specialized Go AIs is unclear

**Quality:**

3

**Strengths And Weaknesses:**

Strength:
- Novel and ambitious application: Attempting to bridge domain expertise with general LLM capabilities is an interesting research direction
- Comprehensive experimental design: The paper includes thorough ablation studies, multiple baselines, and evaluation across both Go-specific and general reasoning tasks
- Transparent methodology: Detailed description of dataset construction, training procedures, and reward function design
- Strong empirical claims: If validated, achieving professional-level Go performance while maintaining general capabilities would be significant

Limitation:
1. Fundamental Architectural Mismatch
The core premise is questionable: using sequential text encoding ("1.X-Q16 2.O-D16...") for a inherently spatial, visual game like Go. LLMs processing 150+ move sequences cannot realistically maintain spatial board understanding. This encoding forces the model to reconstruct 2D spatial relationships from 1D text, which is fundamentally inefficient.

2. Insufficient Justification for Approach
The paper fails to adequately address why this text-based LLM approach is preferable to existing methods. AlphaGo and successors use direct board representations for good reason. The authors should justify why forcing spatial reasoning through text is beneficial.

3. Questionable Performance Claims
- The dramatic improvement from 60-70% to 88%+ through RL seems implausibly large
- Win rates against specialized Go AIs need independent verification
- The KataGo-Bench-1K benchmark is self-constructed and may not reflect true playing strength
- Human evaluation shows only 55.6% correct explanations, suggesting pattern matching rather than understanding

4. Limited Understanding Analysis
- The paper lacks analysis of what the model actually "understands" about Go. The context curse analysis (Figure 8) shows severe performance degradation as games lengthen, suggesting the model doesn't maintain coherent board state understanding.

5. Evaluation Concerns
- Evaluation primarily on authors' own benchmark
- Limited diversity in opponent strength assessment
- No analysis of actual game quality or strategic understanding
- Self-play evaluation would strengthen claims

---

> ### Author Rebuttal · Authors · 2025-07-27
>
> Thank you for your valuable comments! It is exciting for us to communicate with a reviewer who understands both Go and our work, so we are very willing to provide clarifications for your concerns! We will clarify the questions and the weaknesses in order, and we sincerely hope our responses can address your concerns.
>
> ## **Major Question: other LLMs hallucinate, need more evidence of our work**
>
> We are very willing to provide such data to confidently demonstrate the impressive capabilities of our model! Since links are not allowed here, we choose to include `SGF` format game records and show the output of some step. We play LoGos against Golaxy (星阵) Giant Elephant (星巨象) model, which is a widely acknowledged pro-level model. As mentioned in our main text Figure 1, our model achieves a **55% win rate** against this professional Go model (in 60+ games).
>
> ### Game Record 1(LoGos loses):
>
> SGF:
> ```
> (;CA[utf8]AP[golaxy]PW[LoGos]PB[星巨象]WR[Unknown]BR[星阵1星]HA[0]RE[黑中盘胜]DT[25.07.26]KM[7.5]SZ[19]ST[0]RU[chinese];B[pd];W[dd];B[pq];W[dq];B[co];W[cp];B[do];W[fq];B[dk];W[qo];B[op];W[ql];B[cf];W[fc];B[cd];W[cc];B[bc];W[ce];B[bd];W[df];B[be];W[de];B[cg];W[nc];B[qf];W[pc];B[qc];W[qb];B[oc];W[pb];B[od];W[qd];B[rc];W[rd];B[rb];W[ra];B[ob];W[sb];B[kc];W[ic];B[qj];W[rq];B[kq];W[pi];B[pj];W[pg];B[pf];W[ni];B[oj];W[ng];B[qg];W[me];B[mc];W[ld];B[lc];W[je];B[eh];W[fj];B[ej];W[fh];B[fg];W[fi];B[eg];W[el];B[dl];W[gg];B[gf];W[hg];B[fe];W[cb];B[mk];W[lj];B[lk];W[kj];B[kk];W[jk];B[jl];W[il];B[jm];W[iq];B[rr];W[qq];B[qr];W[rk];B[rj];W[pn];B[rm];W[rl])
> ```
>
> Model responses:
> - **Step 2:**
> ```
> 啊，黑棋第一手选择了Q16星位，一个非常经典开局。作为白棋，我们需要好好思考如何应对这个局面。
>
> (轻敲棋盘)让我来分析几种常见的应对方案：
>
> 1. 白棋D16星位对角
> 这是最经典的对角星布局，后续可能形成：
> 2.O-D16
> 3.X-Q3
> 4.O-D3
> ...
> 这种变化下白棋胜率52.3%，相当均衡。这种对角星布局就像两位武林高手各自占据一个角落，各展其能。
> (content omitted due to space constraints)
> ```
>
> - **Step 28:**
> ```
> <reasoning>让我们仔细分析当前的局面。从全局来看，黑棋第27手R17的挡是必然的一手，既保护了自身利益，又对白棋Q17的小尖形成制约。现在轮到白棋，我们需要寻找既能扩张势力又能限制黑棋发展的好点。
>
> 右上角的战斗尤为关键。白棋Q17的小尖已经对黑棋形成压力，此时若在R18的扳是绝佳的反击点：
> 1. 直接对黑棋右上角施加压力
> 2. 与白棋其他子力形成良好配合
> 3. 为后续在右上区域的发展奠定基础
> 4. 同时兼顾右下角的势力发展
>
> 具体变化推演：
> 28.O-R18的扳（妙手！）
> 29.X-P17的打吃（黑棋的反击）
> 30.O-Q18粘住（白棋加固自身）
> 31.X-P16粘住（黑棋补断）
> 32.O-P18粘住（白棋完全安定）
> 33.X-O16长（黑棋的强手）
> 34.O-N17跳（白棋轻松跳出）
>
> 这个变化下白棋胜率53.3%，是一个积极的态势。相比之下，如果选择E9的小飞，虽然能扩张左边，但会放任黑棋在右上角加固，胜率仅50.9%，不够理想。
>
> 特别值得一提的是，R18的扳具有"一石三鸟"的效果：
> - 限制黑棋扩张
> - 巩固自身阵地
> - 为后续作战创造支点
>
> 对手如果应对不当，我们甚至可以在右上角形成"铁壁"般的厚势。不过要提醒自己"不得贪胜"，稳扎稳打才是取胜之道。
> </reasoning>
> <answer>
> {下一步颜色:白}
> {下一步位置:R18}
> {下一步胜率:53.3%}
> </answer>
> ```
>
> - **Step 80:**
> ```
> <reasoning>让我们仔细分析当前局面。这盘棋已经进行到第79手，黑棋刚刚在K7长了一手，这步棋很有意思啊。从整体格局来看，双方在四个角部都形成了星位布局，但战斗主要集中在中央和右侧。
>
> 黑棋这手K7长显示了强烈的进攻意图。通常在这个阶段，黑棋会选择在J3拆边这样稳健的下法，但对手显然不甘于平淡。这手K7长让棋局变得更加复杂有趣了。
>
> 现在轮到白棋的第80手，我认为有几个值得考虑的选择：
>
> 1. J9扳（胜率49.7%）
> 这步棋很直接，但似乎有些操之过急。变化可能是：
> 80.O-J9扳（直接破坏黑棋形状）
> 81.X-Q6点三线（黑棋依然会选择点三线）
> 这个变化下白棋稍显被动，胜率49.7%，不太推荐。
>
> 2. J3拆边（胜率50.5%）
> 如果选择这个变化：
> 80.O-J3拆边（拆边是经典的大场）
> 81.X-Q6点三线（黑棋点三线，白棋拆边，形成经典对抗）
> 82.O-R6挡（黑棋挡下是必然）
> 83.X-J9扳（黑棋扳住，白棋在下方有发展空间）
> 84.O-K10长（白棋长出，形成更稳固的形状）
> 这个变化下，白棋能保持微弱的优势，胜率50.5%，棋很细啊。
>
> （摇头）其他选择如直接打入黑棋阵地，在当前局面下都显得不够明智。毕竟，黑棋在中央已经形成了一定势力，贸然深入可能会陷入被动。
>
> 综合来看，J3拆边确实是当前的最佳选择。这步棋既符合"入界宜缓"的原则，又能为后续战斗创造有利条件。对手如果应对不当，我们还能获得更大优势呢！
> (content omitted due to space constraints)
> ```
>
> ### Game Record 2(LoGos wins):
>
> SGF (only provide part of the records due to space constraints):
> ```
> (;CA[utf8]AP[golaxy]PW[LoGos]PB[星巨象]BR[星阵1星]HA[0]RE[W+R]DT[2025-07-2613:39:32]KM[7.5]SZ[19]ST[0]RU[chinese];B[pd];W[dp];B[pq];W[dd];B[qk];W[pp];B[qp];W[oq];B[op];W[po];B[nq];W[or];B[qq];W[nr];B[oo];W[qo];B[ro];W[rn];B[rp];W[on];B[mq];W[lq];B[mr];W[np];B[no];W[mp];B[nn];W[lr];B[om];W[qc];B[qd];W[pc];B[od];W[nb];B[cq];W[dq];B[cp];W[co];B[bo];W[bn];B[cn];W[do];B[bm];W[bp];B[an];W[bq];B[cf];W[em];B[fc];W[fd];B[gd];W[fe];B[dc];W[cc];B[ec];W[cd];B[dg];W[cj];B[fg];W[eh];B[eg];W[cl];B[be];W[ge])
> ```
>
> Next, we are willing to discuss the **poor performance of other LLMs** you mentioned. We believe the reason is that the task of next step prediction is essentially non-existent in pre-training corpora, making it a completely **out-of-distribution task**. Therefore, general LLMs without specialized training can only hallucinate and randomly guess an output.
>
> As a comparison, from the LoGos outputs provided, you can clearly see that although the model exhibits some hallucination in terminology and analysis, it does **perform reasoning, analysis, and prediction regarding the game situation.** This is significantly different from the random responses of all other LLMs.
>
> Regarding "open-source the dataset, or at least provide some samples," we have already committed in our paper to **open-source all useful information**, including the complete training dataset, training methodology, model checkpoints, and evaluation benchmarks. We firmly believe our model represents **the first genuine step from AlphaGo toward human-understandable Go agents**, and we are looking forward to advancing this work.
>
> ## **Question 1: Why use text encoding for Go when spatial representations are more natural?**
>
> Thank you for this insightful question! After finishing this paper, we spent considerable time researching this issue and ultimately found an effective solution to the context curse problem - **a way to model 2D information through text.**
>
> First, we choose text encoding because we wanted to **elegantly perform Go task reasoning through general LLMs using natural language**. While spatial representation methods are effective, such methods deviate from the original intention of solving specialized problems through natural language. Such models ultimately **remain specialized models for single professional tasks** and cannot output human-understandable reasoning processes. Therefore, we choose to encode Go using text, attempting to combine the powerful general reasoning capabilities of LLMs with Go expertise.
>
> However, as you mentioned, using only move lists presented difficulties for the model's understanding of 2D board states. Through experimentation, we find a new approach: while providing move lists, we also **render the Go board as a 19x19 2D array**, in which 1 represents black stones, -1 represents white stones, and 0 represents empty positions. With the new input format, the model's "context curse" problem is greatly alleviated, achieving **90%+ accuracy in long games(200+ moves)**.
>
> Due to space constraints, please refer to our response to ***reviewer AjQt's Question 1*** for specific ***rendering details and experimental results***.
>
> ## **Question 2: How do you distinguish between learning Go principles versus memorizing patterns from training data?**
>
> This is a very interesting question. In fact, we believe the model definitely memorizes patterns from training data. However, since the number of possible variations (\~250^150) is **significantly larger** than that of positions training data could contain (\~10M), our model necessarily needs to generalize the learned patterns to unseen scenarios. Therefore, given that the model achieves professional-level Go performance, we believe it has **successfully generalized the learned patterns** and acquired correct Go principles.
>
> ## **Question 3: Share actual conversation histories & How does the model handle complex spatial relationships?**
>
> Thank you for your insightful question! In the **Major Question section**, we have already provided some conversation history. The provided responses show that LoGos can make judgments and reasoning based on the game situation quite well.
>
> Additionally, here is an interesting example:
> ```
> CN: 让我们仔细审视当前的局面。这盘棋已经进行到第192手，白棋刚刚在K15提劫，这已经是本局第7次在这个关键点提劫了。看来对手对这个劫争相当执着啊。
>
> Tranlated vers: Let's carefully examine the current situation. This game has reached move 192, with White just taking the ko at K15, which is already the 7th time this point has been captured in this game. It seems the opponent is quite persistent about this ko fight.
> ```
>
> Here the model correctly identifies from the move list that **a ko fight** is occurring at position K15, correctly counted the occurrences, and output such a description.
>
> ## **Question 4: How does performance vary across different board sizes or rule variants not seen in training?**
>
> Thank you for your helpful question! In fact, we conduct RL experiments **on 9x9 games without cold start** and find that the model still cannot directly learn Go strategies. Although we have not conducted complete cold start + RL process on 9x9 or other board sizes, we believe that since 19x19 task is proven to be learnable, reproducing this performance on smaller board sizes should be expected.
>
> As for rules, we use Chinese rules, therefore when switching to other rules, the model's accuracy would shift, requiring additional fine-tuning.
>
> ## **Weakness 1&2**
> Please refer to Q1.
>
> ## **Weakness 3**
>
> ### W3.1 improvement seems implausible
> This is an insightful question! We believe the RL stage is divided into two parts. First, after cold start, the model already achieves nearly 80% accuracy using **template prediction**, indicating that it has learned Go knowledge. Therefore, the first stage of RL is **transferring knowledge** learned from structured data to long COT paradigm reasoning. Next, after the model reaches parity with template prediction performance, it continues **self-exploration under reasoning paradigm**, ultimately improving to 88%+.
>
> ### W3.2&W3.3&W3.4
> Please refer to Major Question and Q1.
>
> ## **Weakness 4**
> Please refer to Major Question. We will add more model outputs to the appendix.
>
> ## **Weakness 5: Evaluation Concerns**
>
> In addition to the Elo analysis mentioned in Q1, we will add more analysis of model outputs in the appendix, including inviting more Go players to help us conduct human evaluation.
>
> ## **Summary**
> Finally, thank you once again for your time and effort during the review process! We are very pleased to have a reviewer who is familiar with Go, and therefore we hope our clarifications can address your concerns. We are also very willing to engage in more discussions with you during the Discussion phase!

---

> ### Author Response · Authors · 2025-08-05
>
> Dear Reviewer Dp9f,
>
> Thank you once again for your thoughtful review and the time you dedicated during the review process! We would like to know your thoughts on the clarifications we provided in our response, and we are very willing to continue discussing any remaining concerns you may have.
>
> Thank you for your consideration.
>
> Best regards,
>
> The Authors

---

> > ### Comment · Reviewer_Dp9f · 2025-08-05
> >
> > Thank you for the comprehensive response and for providing concrete game records and model outputs. I appreciate the detailed engagement with my questions and your commitment to open-sourcing the work.
> > While I value the provided SGF records and conversation samples, several fundamental issues remain unresolved:
> > 1. Incomplete Technical Solution: You mention achieving "90%+ accuracy in long games" through 2D array rendering, but the critical technical details are deferred to another reviewer's response. Without seeing this approach, I cannot evaluate whether it truly addresses the architectural mismatch between text encoding and spatial reasoning that I raised.
> > 2. Fundamental Approach Justification: Your explanation for choosing text encoding ("to elegantly perform Go task reasoning through general LLMs") doesn't demonstrate concrete advantages over existing spatial representation methods. The circular reasoning—assuming LLM approaches are preferable without proving superiority—persists.
> > 3. Understanding vs. Pattern Matching: While you acknowledge the model "definitely memorizes patterns," the evidence for genuine Go understanding versus sophisticated pattern matching remains unconvincing, particularly given the 55.6% explanation accuracy in human evaluation.
> >
> > My skepticism is grounded in substantial research evidence that LLMs struggle significantly with long sequential data, exhibiting the well-documented "lost in the middle" phenomenon where performance is often highest when relevant information occurs at the beginning or end of the input context, and significantly degrades when models must access relevant information in the middle of long contexts" [1]. This fundamental limitation is particularly concerning for Go, where understanding board state requires processing 150+ move sequences.
> >
> > Given these persistent concerns about the fundamental approach, well-documented limitations of LLMs with long sequential data, and incomplete presentation of key technical solutions, I maintain my current rating. However, I strongly encourage you to complete this work with the missing technical details and independent evaluation, especially open-sourcing the dataset. I remain unconvinced by a research approach that contradicts established engineering knowledge about LLMs' poor performance on long sequential data processing tasks.
> >
> > [1]: https://aclanthology.org/2024.tacl-1.9/

---

> > > ### Author Response · Authors · 2025-08-06
> > > **Response to Reviewer Dp9f (2/3)**
> > >
> > > ## **Issue 3: Understanding vs. Pattern Matching**
> > >
> > > Thank you for this feedback! We need to clarify that the 55.6% accuracy refers to cases where the model's reasoning **completely aligns with human players' thinking process** (i.e., terminology accuracy, the moves and variations). The proportion of cases where the model's predicted next move matches human preferences is **218/226 = 96.5%**.
> > >
> > > This phenomenon occurs because, during GRPO (self exploration), reward is designed **only based on move accuracy and predicted win rate**, while **without supervision on terminology accuracy**. Therefore, the model's reasoning process is not strictly aligned with human preferences, leading to cases where the model outputs **incorrect terminology (such as confusing "diagonal" and "knight's move")**, which are marked as ambiguous or incorrect explanations by our invited human players.
> > >
> > > As shown in the provided cases, the model's understanding of the position is truly based on specific game state, and its predicted subsequent variations are reasonable predictions for the current position. Therefore, we believe the model **has indeed developed understanding of the position**. In our response to Q3, we can clearly see the model's interesting understanding of the position:
> > >
> > > ```
> > > Let's carefully examine the current situation. This game has reached move 192, with White just taking the ko at K15, which is already the 7th time this point has been captured in this game. It seems the opponent is quite persistent about this ko fight.
> > > ```
> > >
> > > Here the model correctly identifies from the move list that **a ko fight is occurring at position K15**, correctly counts the occurrences, and outputs such a description. We believe this is strong evidence that the model correctly understands the game state of Go.
> > >
> > > We sincerely hope the above response addresses your concerns!
> > >
> > > ### **Additional Information**
> > > Additionally, here is a translated version of the model response shown in rebuttal:
> > >
> > > > < reasoning >
> > > Let's carefully analyze the current position. From a global perspective, Black's 27th move at R17 is a necessary **block** that both protects Black's own interests and constrains White's **diagonal** at Q17. Now it's White's turn, and we need to find a good move that can both expand influence and limit Black's development.
> > > >
> > > > The battle in the upper right corner is particularly critical. White's diagonal at Q17 has already put pressure on Black. At this point, a **hane** at R18 would be an excellent counter-attacking point:
> > > >
> > > > 1. Directly applies pressure to Black's upper right corner
> > > > 2. Forms good coordination with White's other forces
> > > > 3. Lays the foundation for future development in the upper right area
> > > > 4. Simultaneously considers the development of influence in the lower right corner
> > > >
> > > > Specific variation analysis:
> > > >
> > > > 28. O-R18 **hane** (brilliant move!)
> > > > 29. X-P17 **atari** (Black's counter-attack)
> > > > 30. O-Q18 **tsugi** (White strengthens position)
> > > > 31. X-P16 **tsugi** (Black fills the cut)
> > > > 32. O-P18 **tsugi** (White completely settles)
> > > > 33. X-O16 **stretch** (Black's strong move)
> > > > 34. O-N17 **jump** (White easily escapes)
> > > >
> > > > Under this variation, White's winning rate is 53.3%, which is a positive situation. In comparison, if we choose the **knight's move** at E9, while it can expand the left side, it would allow Black to strengthen the upper right corner, with a winning rate of only 50.9%, which is not ideal.
> > > >
> > > > It's worth mentioning that the hane at R18 has a "killing three birds with one stone" effect:
> > > > - Limits Black's expansion
> > > > - Consolidates White's own territory
> > > > - Creates a foothold for future battles
> > > >
> > > > If the opponent responds poorly, we can even form an iron wall of thickness in the upper right corner. However, we must remind ourselves "don't be greedy for victory" - steady and solid play is the path to winning.
> > > >
> > > > < / reasoning >

---

> > > ### Author Response · Authors · 2025-08-08
> > >
> > > Dear Reviewer Dp9f,
> > >
> > > Thank you once again for your detailed review and feedback! We greatly appreciate the time you've invested in reviewing our work.
> > >
> > > Given that **the discussion period is nearing its end**, we would like to check if our responses have addressed your concerns, or if there are any remaining questions to be discussed. We are very happy to continue the discussion and provide any additional information that would be helpful.
> > >
> > > Thank you again for your valuable input.
> > >
> > > Best regards,
> > >
> > > The Authors

---

> ### Author Response · Authors · 2025-08-06
> **Response to Reviewer Dp9f (1/3)**
>
> Thank you for your detailed and thoughtful feedback! We are wiiling to provide the following clarifications for the issues you mentioned.
>
> ## **Issue 1: Technical details (90%+ accuracy in long games) are deferred to another reviewer's response**
>
> We apologize for the inconvenience. Due to space constraints (SGF files occupied a large number of characters), when responding to Q1, we referenced from our response to Reviewer AjQt. We would be happy to also present the rendering method and experimental results mentioned here.
>
> Under such a move list:
> ```
> 1.X-D16 2.O-D4 3.X-Q4 4.O-Q16 5.X-O17 6.O-R14 7.X-C3 8.O-D3 9.X-C4 10.O-D5 11.X-B6 12.O-R6
> ```
>
> The rendered 2D array is: (1 represents black, -1 represents white, and 0 represents empty positions)
> ```
> [[0, 0, 0, 0, 0, 0, 0, 0, 0, 0, 0, 0, 0, 0, 0, 0, 0, 0, 0],
>  [0, 0, 0, 0, 0, 0, 0, 0, 0, 0, 0, 0, 0, 0, 0, 0, 0, 0, 0],
>  [0, 0, 0, 0, 0, 0, 0, 0, 0, 0, 0, 0, 0, 1, 0, 0, 0, 0, 0],
>  [0, 0, 0, 1, 0, 0, 0, 0, 0, 0, 0, 0, 0, 0, 0, -1, 0, 0, 0],
>  [0, 0, 0, 0, 0, 0, 0, 0, 0, 0, 0, 0, 0, 0, 0, 0, 0, 0, 0],
>  [0, 0, 0, 0, 0, 0, 0, 0, 0, 0, 0, 0, 0, 0, 0, 0, -1, 0, 0],
>  [0, 0, 0, 0, 0, 0, 0, 0, 0, 0, 0, 0, 0, 0, 0, 0, 0, 0, 0],
>  [0, 0, 0, 0, 0, 0, 0, 0, 0, 0, 0, 0, 0, 0, 0, 0, 0, 0, 0],
>  [0, 0, 0, 0, 0, 0, 0, 0, 0, 0, 0, 0, 0, 0, 0, 0, 0, 0, 0],
>  [0, 0, 0, 0, 0, 0, 0, 0, 0, 0, 0, 0, 0, 0, 0, 0, 0, 0, 0],
>  [0, 0, 0, 0, 0, 0, 0, 0, 0, 0, 0, 0, 0, 0, 0, 0, 0, 0, 0],
>  [0, 0, 0, 0, 0, 0, 0, 0, 0, 0, 0, 0, 0, 0, 0, 0, 0, 0, 0],
>  [0, 0, 0, 0, 0, 0, 0, 0, 0, 0, 0, 0, 0, 0, 0, 0, 0, 0, 0],
>  [0, 1, 0, 0, 0, 0, 0, 0, 0, 0, 0, 0, 0, 0, 0, 0, -1, 0, 0],
>  [0, 0, 0, -1, 0, 0, 0, 0, 0, 0, 0, 0, 0, 0, 0, 0, 0, 0, 0],
>  [0, 0, 1, -1, 0, 0, 0, 0, 0, 0, 0, 0, 0, 0, 0, 1, 0, 0, 0],
>  [0, 0, 1, -1, 0, 0, 0, 0, 0, 0, 0, 0, 0, 0, 0, 0, 0, 0, 0],
>  [0, 0, 0, 0, 0, 0, 0, 0, 0, 0, 0, 0, 0, 0, 0, 0, 0, 0, 0],
>  [0, 0, 0, 0, 0, 0, 0, 0, 0, 0, 0, 0, 0, 0, 0, 0, 0, 0, 0]]
> ```
>
> Under the new setting, the model's input query contains **both move list information and actual board state information**. After following the same experimental procedure (cold start + RL), we are excited to see that the 'context curse' problem is greatly alleviated:
>
> | Model | 1-50 | 51-100 | 101-150 | 151-200 | 200+ |
> |-------|------|--------|---------|---------|------|
> | **Former Models** | | | | | |
> | Original-RL_Step-1600 | 99.1 | 95.0 | 87.7 | 78.0 | 74.4 |
> | Original-RL_Step-1000 | 96.1 | 91.3 | 81.3 | 78.0 | 64.1 |
> | Original-RL_Step-500 | 86.1 | 75.2 | 68.0 | 59.3 | 33.3 |
> | **New Models** | | | | | |
> | Addboard-RL_Step-1400 | **97.3** | 94.3 | **91.1** | **92.3** | 89.7 |
> | Addboard-RL_Step-1200 | 96.1 | **94.6** | 89.2 | 85.7 | **92.3** |
>
> We find that the models with updated input information demonstrates significantly **more accurate understanding of board states**, maintaining high prediction accuracy even under 200+ moves. Therefore, we believe this new method effectively addresses the context curse challenge.
>
> We hope the above clarifications address your concerns!
>
> ## **Issue 2: Explanation for choosing text encoding ("to elegantly perform Go task reasoning through general LLMs") doesn't demonstrate concrete advantages over existing spatial representation methods.**
>
> This is an interesting question. Our reasoning is as follows. Firstly, our motivation is to enable general LLMs to solve Go reasoning tasks, so using natural language input is a more natural approach.
>
> While models like AlphaGo effectively use spatial representation, they are specialized Go models. **Their move selection process (e.g., MCTS) is difficult for human players to understand**. For general LLMs (which can generate human-understandable COT reasoning), spatial representation may be helpful, but it requires additional adjustments (like modifying tokenizers). In contrast, our text encoding approach directly leverages the model's instruction following capability, making it more direct and natural.
>
> We **do not claim that text encoding is better**, but since text encoding is more aligned with LLM usage scenarios and its effectiveness has been proven to work, we believe that **using the text encoding approach is reasonable.**
>
> We wish that the above clarifications address your concerns!

---

> ### Author Response · Authors · 2025-08-06
> **Response to Reviewer Dp9f (3/3)**
>
> ## **Regarding "Lost in the middle"**
>
> This is a very thoughtful question! Although we believe that the 2D-array rendering method we provided in Issue 1 and rebuttal Q1 **can effectively address this issue**, we are still happy to discuss why **previous models** also demonstrated strong capabilities.
>
> We believe that "lost in the middle" is not a significant constraint for Go tasks. This is because the input for Go tasks **does not reach the level of long context** - with an average prompt length of around 800 tokens, and given the base model's (Qwen2.5-7B-Instruct) 32K context window, this should not be considered a long-context task. "Lost in the middle" typically refers to information retrieval tasks in **much longer text scenarios**.
>
> The challenge with using move list text encoding for Go tasks is that in longer game sequences, the model has more difficulty **understanding the overall board state**, making it **more likely to respond locally** rather than finding the best next move for the entire position. We provide detailed demonstrations of this phenomenon in Appendix C. From a Go perspective, this is somewhat ***similar to the difficulties humans encounter when playing mekura-go (blindfold go)***.
>
> Therefore, to address this issue, we explore the 2D array rendering method, allowing the model to directly access position states. This method achieves surprisingly good results. **Since the model maintains high prediction accuracy even with larger game sequences, we believe this issue can be resolved.**
>
> ## **Summary**
>
> In short, we believe your concerns are very thoughtful, but given LoGos' demonstrated capabilities, **we are confident that LLMs can effectively handle Go tasks**. We hope the experiments and explanations we provided in our rebuttal and comments can demonstrate this point to you.
>
> We sincerely hope our response addresses your concerns, and if you have other concerns about our clarification, **we are also happy to continue engaging in discussion!**

---

### Official Review · Reviewer_AjQt · 2025-07-02

**Clarity:** 2
**Significance:** 3
**Originality:** 3
**Rating:** 5
**Confidence:** 4

**Summary:**

This paper introduces LoGos, a large language model (LLM) that is specialized in playing the game of Go. The model is trained on a novel dataset that has been constructed based on real, annotated game states and Go commentary. A novel training paradigm, using the aforementioned dataset and chain-of-thought (CoT) reasoning data from a variety of domains for fine-tuning, is applied. Reinforcement learning is applied to improve the model’s next-move prediction performance, using a reward function that encourages the model to output the correct next move, given a game state, alongside the win rate estimation for the predicted move. LoGos outperforms other LLMs and a competitive AI system, KataGo, in accuracy on predicting the next move given a game state, while also maintaining general capabilities on math, reasoning, and coding benchmarks. According to human evaluators, LoGos produces accurate move predictions 96.5% of the time, but the explanations given in natural language are correct only 55.6% of the time.

**Questions:**

- Do the authors have suggestions for future work that could address the ‘context curse’ issue?
- The proposed benchmark includes a measure of accuracy for next-move prediction, given game state, but is there a way to assess LoGos’ overall win rate in complete games?
- Can the authors provide a reference for: "Go contains more than 600 terms" (p. 8)?

**Ethical Concerns:**

["Major Concern: Data privacy, copyright, and consent"]

**Final Justification:**

The authors have addressed many of the initial concerns I had about the paper. I raised my score from 4 to 5.

**Limitations:**

The aforementioned concerns regarding the data sourcing should be addressed.

**Quality:**

2

**Strengths And Weaknesses:**

Strengths:
Integrating natural language explanations into Go-playing AI systems could be very a valuable entertainment and educational resource for the Go-playing community. The proposed model achieves admirable performance on a benchmark constructed to measure the accuracy of next-step prediction, given a game state. This paper represents a promising direction in Go AI research.

Weaknesses:
Performance concerns: The following is acknowledged in the paper, but remains a considerable concern for the utility of the proposed model. Over the course of a game of Go, the context contained in the LLM input becomes increasingly long, and performance degrades. Additionally, the natural language explanations, which could be a core highlight of the proposed system, have accuracy issues.

Exposition: The description of the game of Go in Section 2.1 could be improved. There is not explicit information about the number of players, nor what constitutes a winning state for the game. It may also be helpful to explain basic mechanics of the game, like “capture”. The paper is not very accessible to people who don’t already know how to play Go.

Reproducibility concerns: There is insufficient information about the source of the Go commentary data. It would be helpful to have information about the properties of the data, e.g. average length of comments, language(s) included, and whether the commentary is automatically transcribed from video/audio data or inherently text-based. This data is crucial to the approach presented, so it seems important for the audience to be able to assess its provenance and quality. Additionally, there are potential ethical issues: even though the data is ostensibly ‘public,’ do the authors have permission to collect and redistribute it?

---

> ### Author Rebuttal · Authors · 2025-07-27
>
> Thank you for your valuable comments! We sincerely appreciate your review, detailed comments, and valuable suggestions, and are pleased to address your concerns and provide the following clarifications. We will clarify the questions and the weaknesses in order, and we sincerely hope our responses can address your concerns.
>
> ## **Question 1: Do the authors have suggestions for future work that could address the 'context curse' issue?**
>
> Thank you for your insightful question! In fact, during the period after submission, we conducted extensive experiments to address this issue and ultimately **found a quite effective solution**. Since Go is actually a game played on a 2D plane, we believe it is necessary for the model to **see the actual state of the board**. Therefore, while providing the move list to the model, we also render the Go board as a **19×19 2D array** and provide it to models in the query.
>
> In this 2D array, **1 represents black, -1 represents white, and 0 represents positions where no stones have been placed yet**. Our rendering strategy complies with all Go rules (such as the rule of removing stones from the board).
>
> For example, given the following move list:
> ```
> 1.X-D16 2.O-D4 3.X-Q4 4.O-Q16 5.X-O17 6.O-R14 7.X-C3 8.O-D3 9.X-C4 10.O-D5 11.X-B6 12.O-R6
> ```
>
> The rendered 2D array is:
> ```
> [[0, 0, 0, 0, 0, 0, 0, 0, 0, 0, 0, 0, 0, 0, 0, 0, 0, 0, 0],
>  [0, 0, 0, 0, 0, 0, 0, 0, 0, 0, 0, 0, 0, 0, 0, 0, 0, 0, 0],
>  [0, 0, 0, 0, 0, 0, 0, 0, 0, 0, 0, 0, 0, 1, 0, 0, 0, 0, 0],
>  [0, 0, 0, 1, 0, 0, 0, 0, 0, 0, 0, 0, 0, 0, 0, -1, 0, 0, 0],
>  [0, 0, 0, 0, 0, 0, 0, 0, 0, 0, 0, 0, 0, 0, 0, 0, 0, 0, 0],
>  [0, 0, 0, 0, 0, 0, 0, 0, 0, 0, 0, 0, 0, 0, 0, 0, -1, 0, 0],
>  [0, 0, 0, 0, 0, 0, 0, 0, 0, 0, 0, 0, 0, 0, 0, 0, 0, 0, 0],
>  [0, 0, 0, 0, 0, 0, 0, 0, 0, 0, 0, 0, 0, 0, 0, 0, 0, 0, 0],
>  [0, 0, 0, 0, 0, 0, 0, 0, 0, 0, 0, 0, 0, 0, 0, 0, 0, 0, 0],
>  [0, 0, 0, 0, 0, 0, 0, 0, 0, 0, 0, 0, 0, 0, 0, 0, 0, 0, 0],
>  [0, 0, 0, 0, 0, 0, 0, 0, 0, 0, 0, 0, 0, 0, 0, 0, 0, 0, 0],
>  [0, 0, 0, 0, 0, 0, 0, 0, 0, 0, 0, 0, 0, 0, 0, 0, 0, 0, 0],
>  [0, 0, 0, 0, 0, 0, 0, 0, 0, 0, 0, 0, 0, 0, 0, 0, 0, 0, 0],
>  [0, 1, 0, 0, 0, 0, 0, 0, 0, 0, 0, 0, 0, 0, 0, 0, -1, 0, 0],
>  [0, 0, 0, -1, 0, 0, 0, 0, 0, 0, 0, 0, 0, 0, 0, 0, 0, 0, 0],
>  [0, 0, 1, -1, 0, 0, 0, 0, 0, 0, 0, 0, 0, 0, 0, 1, 0, 0, 0],
>  [0, 0, 1, -1, 0, 0, 0, 0, 0, 0, 0, 0, 0, 0, 0, 0, 0, 0, 0],
>  [0, 0, 0, 0, 0, 0, 0, 0, 0, 0, 0, 0, 0, 0, 0, 0, 0, 0, 0],
>  [0, 0, 0, 0, 0, 0, 0, 0, 0, 0, 0, 0, 0, 0, 0, 0, 0, 0, 0]]
> ```
>
> Under the new setting, the model's input query contains **both move list information and actual board state information**.
>
> After following the same experimental procedure (cold start + RL), we are excited to see that using the new query, the model's 'context curse' problem is greatly alleviated. The table below shows the evaluation results on KataGo-Bench-1K:
>
> | Model | 1-50 | 51-100 | 101-150 | 151-200 | 200+ |
> |-------|------|--------|---------|---------|------|
> | **Former Models** | | | | | |
> | Original-RL_Step-1600 | 99.1 | 95.0 | 87.7 | 78.0 | 74.4 |
> | Original-RL_Step-1000 | 96.1 | 91.3 | 81.3 | 78.0 | 64.1 |
> | Original-RL_Step-500 | 86.1 | 75.2 | 68.0 | 59.3 | 33.3 |
> | **New Models** | | | | | |
> | Addboard-RL_Step-1400 | **97.3** | 94.3 | **91.1** | **92.3** | 89.7 |
> | Addboard-RL_Step-1200 | 96.1 | **94.6** | 89.2 | 85.7 | **92.3** |
>
> We find that the models with updated input information demonstrates significantly **more accurate understanding of board states**, maintaining high prediction accuracy even under game sequences with 200+ moves. Therefore, we believe this new method effectively addresses the context curse challenge. We will supplement this interesting experimental result to the paper and provide corresponding discussions.
>
> We hope the above clarifications address your concerns!
>
> ## **Question 2: Is there a way to assess LoGos’ overall win rate in complete games?**
>
> This is a very valuable question. From our perspective, the assessment of overall win rate in complete games can be **correlated with the game's final win rate**. In fact, to verify the fairness of the KataGo-Bench-1K evaluation set, we also sample several checkpoints for mutual gameplays and calculate the Elo scores of different checkpoints using the following algorithm. This part of the experiment is recorded in Appendix C.
>
> In our designed Elo algorithm (following similar designs in [1-2]), for a certain player A, the score update formula is:
>
> $$\text{Elo}(A) _ {\text{new}} = \text{Elo}(A) _ {\text{old}} + K \cdot (S _ A - E _ A)$$
>
> Where:
> - $\text{Elo}(A)_{\text{new}}$ is the updated Elo rating of player A after the match
> - $\text{Elo}(A)_{\text{old}}$ is the previous Elo rating of player A
> - $K$ is the weight coefficient that determines how much a single game will impact the rating (set to 32)
> - $S_A$ is the actual score of player A in the match (1 for a win, 0.5 for a draw, 0 for a loss)
> - $E_A$ is the expected score of player A, calculated as:
>
> $$E_A = \frac{1}{1 + 10^{(\text{Elo}(B) - \text{Elo}(A))/400}}$$
>
> We conduct pairwise matches with 50 games each, and calculated the Elo scores of various checkpoints (initial score of 1500) using the above formula. The table below shows the Elo scores and KataGo-Bench-1K scores of different checkpoints:
>
> | Model Name | Elo Score | KataGo-Bench-1K |
> |------------|-----------|-----------------|
> | GRPO_1600steps(32B) | 2139.25 | 89.6 |
> | GRPO_1500steps(32B) | 2130.93 | 88.6 |
> | KataGO-HumanSL-9d | 1895.64 | 87.8 |
> | GRPO_1000steps(32B) | 1851.83 | 85.0 |
> | GRPO_800steps(32B) | 1729.03 | 84.4 |
> | GRPO_200steps(7B) | 1515.56 | 78.3 |
> | GRPO_600step(32B) | 1513.23 | 83.7 |
> | GRPO_700steps(32B) | 1506.41 | 85.5 |
> | KataGO-HumanSL-7d | 1476.81 | 81.9 |
> | GRPO_900steps(7B) | 1435.78 | 86.7 |
> | GRPO_60steps(32B) | 1153.67 | 72.3 |
> | Initial point(32B) | 1040.59 | 68.18 |
> | KataGo-Human-SL-18k | 879.72 | 67.4 |
>
> We believe that the Elo score here is **an indicator that can "assess LoGos' overall win rate in complete games."**
>
> It is also worth mentioning that we further calculated the correlation coefficient between the KataGo-Bench-1K scores and Elo, obtaining a result of **r=0.92**, which fully demonstrates the **high correlation between KataGo-Bench-1K and the actual game performance of the models**. Therefore, the performance on KataGo-Bench-1K can also well measure the overall win rate in complete games.
>
> ## **Question 3: Can the authors provide a reference for: "Go contains more than 600 terms" (p. 8)?**
>
> Thank you for this rigorous question. The statement about Go **having 600+ terms** comes from the *Chinese-English Dictionary of Weiqi Terms*, which contains 643 bilingual Chinese-English Go terms. We will supplement this information in the appendix.
>
> ## **Weakness 1: The context contained in the LLM input becomes increasingly long, and performance degrades.**
>
> Please refer to our clarification to question 1.
>
> ## **Weakness 2: The description of the game of Go in Section 2.1 could be improved.**
>
> Thank you for your thoughtful suggestion! Considering the page limit constraints, we only briefly introduce our modeling approach for the Go task in Section 2.1. We will add the necessary rule descriptions and the supplementary information you mentioned in the Appendix. Here we can provide some simple explanations for better understanding:
>
> 1. **Go is a two-player game** where one player uses black stones and moves first, while the other player uses white stones. The two players take turns placing stones.
>
> 2. **In Go, each stone must be placed on an intersection point**. Each stone survives if and only if it has at least one "liberty" (breathing space). When a stone is surrounded by the opponent's stones on all adjacent sides, it is considered to have **zero liberties** and is **"captured"**. Typically, a game ends when all stones from both sides are alive, and further moves cannot guarantee survival. At this point, both players agree to end the game and count the total number of their surviving stones plus the enclosed intersection points. The player with the higher total wins. For experienced players, the game can end simply by mutual agreement that all moves are complete and there is no more question about the life and death status of stones on the board, without needing to continue placing stones.
>
> 3. **"Capture" is a professional Go term** that refers to the action of placing a stone that removes all liberties from the opponent's stones, thereby clearing the captured stones from the board. This is one of the most fundamental rules of Go.
>
> We hope the above explanations help you better understand the rules of this fascinating game, and our work as well！
>
> ## **Weakness 3: There is insufficient information about the source of the Go commentary data.**
>
> This is a very helpful suggestion. Our commentary dataset consists primarily of Chinese language data, with lengths ranging from **10-50 tokens**. Our comments are not derived from video/audio data, **all obtained from publicly available or authorized Go game library files** in `SGF` format (a specialized format for storing Go game records). The game states being commented on are mostly from professional players' live games, with commentators being top amateur players or professional players. There are no ethical or privacy concerns about our commentary data. We will add this information to the appendix.
>
> ## **Summary**
>
> Finally, thank you once again for your time and effort during the review process! We hope that these clarifications address your concerns about the paper, and we look forward to engaging in further discussions with you during the Discussion phase!
>
> ## **References**
>
> [1] Zhang Y, Han X, Li H, et al. Complete Chess Games Enable LLM Become A Chess Master[J]. arXiv preprint arXiv:2501.17186, 2025.
>
> [2] Tian Y, Ma J, Gong Q, et al. Elf opengo: An analysis and open reimplementation of alphazero[C]//International conference on machine learning. PMLR, 2019: 6244-6253.

---

> > ### Comment · Reviewer_AjQt · 2025-08-01
> >
> > Thank you for the comprehensive responses.
> >
> > - Regarding Q1: Rendering the board state as a 2D array indeed seems like a promising remedy for the context collapse problem. Is this a novel idea from the authors, or is there prior work to reference here? In any case, great work with these new results.
> >
> > - Regarding Q2: Appreciate the response. It would be helpful to have some background information about the Elo rating and the rationale for the authors' proposed adaptation to it.
> >
> > - Regarding W3: Thank you for the clarification. Just to make it concrete, is there a particular website or forum from which the data is collected? It will be helpful for reviewers to also verify the terms of use for the sites / data. Also, it will be helpful to add a brief definition of SGF format to the paper.
> >
> > Thanks again.

---

> > > ### Author Response · Authors · 2025-08-02
> > >
> > > Thank you for your detailed feedback! We are pleased that our response address some of your concerns, and we are happy to provide replies to the new questions.
> > >
> > > ## **Regarding Q1:**
> > >
> > > Our method should be novel in the context of Go gameplay, as **no other work has addressed the challenge of general LLM Go gameplay**, let alone solving the context collapse problem. Additionally, to our knowledge, in works using LMs to solve game tasks like Chess(ChessGPT[1], Mastering Board Games by External and Internal Planning with Language Models
> > > [2]), there have been no attempts using similar methods, with the basic input format still being move lists. We believe this is because chess, compared to Go, has significantly **fewer game depths**, so chess tasks may not encounter the difficulties that Go tasks face when game length becomes very long.
> > >
> > > Our inspiration partly comes from works on LLMs handling **puzzle games** (such as KOR-Bench[3], etc.). In these puzzle game tasks, models are provided with a grid and a task designed based on grid information. Our inspiration **mainly stems from how humans play Go**, as Go is typically played based on 2-D board information. Therefore, we believe that rendering the board as a 2D array can also provide useful information for LLMs.
> > >
> > > ## **Regarding Q2:**
> > >
> > > We are glad that our clarification addresses your concern!
> > >
> > > ## **Regarding Q3:**
> > >
> > > Thank you for your question! Due to official reasons, we are not allowed to include links here. Our commentary data comes from some authorized Go platforms, including a well-known Go platform called **Yike Weiqi**. In addition to data from these platforms, in our experiments with 2-D format input(experiments in Q1), we used Go gameplay data from a public GitHub repository (**using GPL-3.0 license**), which only contains player rank information and game moves.
> > >
> > > Regarding the SGF format, we will add a brief introduction in the paper (including common ways of using SGF format data). Here we can provide a simple description of SGF data:
> > >
> > > SGF data is a format used for saving Go game records. At the beginning of SGF data are player information and game rule descriptions. For example, **PW refers to player white**, PB refers to player black, and WR and BR refer to white/black rating, which represents the players' ranks. The first position starting with **B marks the beginning of the game**, where B refers to black's move and W refers to white's move, with the coordinates in `[]` representing positions on the board. SGF uses two letters (representing horizontal and vertical coordinates respectively) to mark positions . SGF is typically used in conjunction with professional Go game record viewing software, such as **Sabaki**.
> > >
> > > We will add the above information to the Appendix.
> > >
> > > ## **Summary**
> > >
> > > In short, we are glad that our clarifications address some of your concerns! Thank you once again for your feedback, and we hope our new replies can resolve your questions.
> > >
> > > ## **References**
> > >
> > > [1] Feng X, Luo Y, Wang Z, et al. Chessgpt: Bridging policy learning and language modeling[J]. Advances in Neural Information Processing Systems, 2023, 36: 7216-7262.
> > >
> > > [2] Schultz J, Adamek J, Jusup M, et al. Mastering board games by external and internal planning with language models[J]. arXiv preprint arXiv:2412.12119, 2024.
> > >
> > > [3] Ma K, Du X, Wang Y, et al. Kor-bench: Benchmarking language models on knowledge-orthogonal reasoning tasks[J]. arXiv preprint arXiv:2410.06526, 2024.

---

### Official Review · Reviewer_eZ1d · 2025-07-03

**Clarity:** 2
**Significance:** 2
**Originality:** 2
**Rating:** 4
**Confidence:** 5

**Summary:**

This paper studies how to incorporate domain-specific knowledge in Go into a large language model while preserving general reasoning capabilities. The authors collected a large-scale next-move prediction Go dataset, formatted with a heuristically generated template, and a moderate-scale commentary is also processed to be part of the training data (give commentary with terminologies in natural language). For training general-purpose LLMs to equip Go knowledge, the authors further propose a two-phrase pipeline (also widely seen in recent RL for LLM reasoning papers) including a mixed cold start fine-tuning and reinforcement learning (GRPO) with KataGo annotations as reward signals.

Comprehensive experiments were made to evaluate both the Go-specific performance and general reasoning competence of the two-phrase trained models. Trained models (with Qwen as base) can attain professional-level Go capabilities and preserve general mathematical reasoning performance (for base models, it can even boost mathematical reasoning).

**Questions:**

What are the Elo scores of the models? What is the performance of Mixed Cold Start (without RL)?

**Ethical Concerns:**

["NO or VERY MINOR ethics concerns only"]

**Final Justification:**

After reading the authors' response, I have decided to increase my rating by one. Although there might be limited technical contributions, I think this paper might be of interest to a general audience at NeurIPS, so I recommend weak acceptance.

**Limitations:**

yes

**Paper Formatting Concerns:**

Table 2 does not satisfy NeurIPS table formatting rule.

**Quality:**

2

**Strengths And Weaknesses:**

#### Strengths
- The paper is well-written and enjoyable to read. In general, it is interesting to see how general-purpose large language models can be turned into professional-level Go models with some curated data. It's an interesting topic and I think Go itself is a good domain for testing these RL bootstrap reasoning methods.

- The collected next-move prediction + commentary dataset is good, and I acknowledge the authors' effort made on this. It would be a good resource for the community to have easy access to Go-specific knowledge in a natural language format.

- Extensive experiments show the effectiveness of the proposed method. It's also good to see how general reasoning performance is preserved. It is also a bit surprising to see that being trained on Go dataset can lead to improvements on mathematical reasoning tasks (even outperforming classic instruction fine-tuning in some cases).

#### Weaknesses
- Limited evaluations. For evaluating Go-specific performance, this paper uses the self-proposed KataGo-Bench. Although there is a correlation analysis between Elo rating and KataGo-Bench, I'm still curious about the Elo score of the trained models. In my memory (correct me if I'm wrong), GPT-3.5/4 already has an Elo score of 1700+ (which is about 85 on KataGo-Bench in Figure 9), But why did closed-source models in Table 1 perform so poorly?

- Limited methodological contributions. It is not surprising that cold-start + GRPO can be applied to LLMs to boost domain-specific reasoning capabilities like Go. I see the main contribution of this paper as the KataGo-Bench and the Expert-level go dataset. The two-phase RL-style training is more like a baseline evaluation of the effectiveness of the proposed dataset rather than a major contribution itself.

---

> ### Author Rebuttal · Authors · 2025-07-27
>
> Thank you for your insightful comments! We sincerely appreciate your review, detailed comments, and valuable suggestions, and are pleased to address your concerns and provide the following clarifications. We will clarify the weaknesses and the questions in order, and we sincerely hope our responses can address your concerns.
>
> ## **Weakness 1: Question about Elo score (closed-source models' performance)**
>
> Thank you for this thoughtful question! Regarding the question about GPT-3.5/4 Elo scores, we must clarify that the "1700+ Elo score" was achieved on **Chess**, **NOT on Go** (Due to official requirements, we cannot provide links, but this information is publicly available through web search).
>
> In our Go gameplay testing, all well-known general-purpose large models (including GPT4, o1, DeepSeek-R1, Gemini, Claude, Doubao, Qwen-Max) perform **far below even beginner level**, with performance essentially equivalent to **randomly making moves on the board**. Therefore, we believe calculating Elo scores for these models is meaningless, since they cannot even finish a game without violating basic rules (like making invalid moves).
>
> **As of the time we write this rebuttal** (July 25, 2025), we can confidently assert that, all known general-purpose large language models, whether closed-source or open-source, **fall far short of Go beginner level** (i.e., the performance of KataGo-HumanSL-18K). Our model is **the first and only general large language model** to achieve professional-level performance in the Go domain.
>
> We sincerely hope that the above clarifications address your concern!
>
> ## **Weakness 2: The two-phase RL-style training (cold-start+RL) is more like a baseline evaluation of the effectiveness of the proposed dataset rather than a major contribution itself.**
>
> Thank you for your valuable question! However, we **beg to differ** that your perspective on our method (limited methodological contributions) has certain limitations. While cold-start + GRPO has indeed become a common practice for reasoning enhancement of LLMs in general domains, most existing works still rely on **distillation data from stronger reasoning models** for their cold-start methods. For example, Fin-R1[1], Med-R1[2], Retool[3], Table-R1[4], Advancing Multimodal Reasoning via Reinforcement Learning with Cold Start[5], etc.
>
> However, for the Go domain, since even the strongest reasoning models cannot obtain correct reasoning paths, the approach of using stronger reasoning models for distillation has completely failed.
>
> To address this, we propose a novel cold start + RL approach. Instead of distilling from strong LLMs, we construct expert-level Go dataset based on heuristic rules **(which is scalable and does not require a generative model)**, and mix it with a long COT general reasoning dataset for cold start training. Next, by adjusting the instructions, we **guide the model to perform Go tasks in the form of long COT reasoning** during the GRPO phase, successfully transferring the Go expertise learned in the cold start stage to the long COT reasoning paradigm.
>
> Thus, we propose a training method that **combines the reasoning capabilities of large language models on general tasks with structured, non-natural language expert knowledge from out-of-distribution (OOD) tasks**, ultimately enabling general LLMs to perform high-quality natural language reasoning on specialized tasks. During the RL phase, we combine the reasoning ability learned from general reasoning data with the Go knowledge acquired from specialized Go dataset, and successfully transfer the expert knowledge into the long COT reasoning paradigm. Compared to tasks that can directly use distillation datasets or high quality natural language corpora for cold start, our method **both provides and proves the effectiveness of a novel practice for specialized domains like Go**, where NL corpora are scarce.
>
> In short, our method differs from existing training paradigms in terms of the **problem being solved**, **the data systhesizing approach for cold-start**, **the purpose of RL**, and the **specific experimental design of RL**. Therefore, our method possesses its own novelty and contributions.
>
> We sincerely hope the above clarification addresses your concerns.
>
> ## **Question 1: What are the Elo scores of the models?**
> Please refer to our clarification to weakness 1.
>
> ## **Question 2: What is the performance of Mixed Cold Start (without RL)?**
> Thank you for this rigorous question. In Figure 6(a), we demonstrate the performance change curve of the model during the GRPO process. At step 0, i.e., before training begins, this value represents the performance immediately after cold-start, where the 7B model achieves **62.63%** and the 32B model achieves **68.18%**. As a comparison, the KataGo-HumanSL-18K model, representing beginner-level performance, performs **67.4%**. This proves that after cold-start alone, the initial point cannot yet perfectly transfer the knowledge learned from the Expert-Level Go Dataset to long CoT paradigm reasoning.
>
> ## **Summary**
>
> Finally, thank you once again for your time and effort during the review process! We hope that these clarifications address your concerns about the paper, and we look forward to engaging in further discussions with you during the Discussion phase!
>
> ## **References**
>
> [1] Liu Z, Guo X, Lou F, et al. Fin-r1: A large language model for financial reasoning through reinforcement learning[J]. arXiv preprint arXiv:2503.16252, 2025.
>
> [2] Lai Y, Zhong J, Li M, et al. Med-r1: Reinforcement learning for generalizable medical reasoning in vision-language models[J]. arXiv preprint arXiv:2503.13939, 2025.
>
> [3] Feng J, Huang S, Qu X, et al. Retool: Reinforcement learning for strategic tool use in llms[J]. arXiv preprint arXiv:2504.11536, 2025.
>
> [4] Yang Z, Chen L, Cohan A, et al. Table-r1: Inference-time scaling for table reasoning[J]. arXiv preprint arXiv:2505.23621, 2025.
>
> [5] Wei L, Li Y, Zheng K, et al. Advancing Multimodal Reasoning via Reinforcement Learning with Cold Start[J]. arXiv preprint arXiv:2505.22334, 2025.

---

> > ### Comment · Reviewer_eZ1d · 2025-08-04
> >
> > Thanks for the authors' response. After reading the rebuttal, I think my concerns were addressed. I have decided to increase my score by one accordingly.

---

> > > ### Author Response · Authors · 2025-08-05
> > >
> > > Thank you for your thoughtful review and for taking the time to carefully evaluate our work! We're pleased that our clarifications addressed your concerns and appreciate your consideration of our responses.

---

### Official Review · Reviewer_vfsx · 2025-07-08

**Clarity:** 3
**Significance:** 2
**Originality:** 2
**Rating:** 3
**Confidence:** 3

**Summary:**

This paper proposes a method for integrating domain-specific expert knowledge into general-purpose LLMs, using the game of Go as a case study. The authors introduce LoGos, a general LLM that achieves professional-level Go play while maintaining strong performance on general reasoning tasks like math and code. The key idea involves a two-phase process: (1) mixed cold start fine-tuning using a combination of domain-specific (Go) structured data and general CoT data; and (2) reinforcement learning via GRPO using a custom reward function designed around move quality and win rate estimation. The authors also construct a large-scale Go dataset and a new evaluation benchmark (KataGo-Bench-1K), releasing both to the community. Experimental results show that LoGos surpasses all existing general LLMs in Go performance and matches or exceeds strong amateur and professional Go models, while still performing competitively on standard general benchmarks.

**Questions:**

1. Could the authors clarify what specific aspects of the training pipeline are novel? How does this approach differ from existing fine-tuning, RLHF or domain-alignment methods?

2. The paper posits that the proposed method can be extended to other specialized domains. Could the authors discuss concrete application areas where similar structured expert knowledge and heuristic rules could be constructed at scale? Are there any early results or experiments in these domains?

3. Does the Go domain offer any general insights that can inform applications in other fields?

**Ethical Concerns:**

["NO or VERY MINOR ethics concerns only"]

**Final Justification:**

After carefully reading the entire discussion between the author and all reviewers, I appreciate the author's effort and have accordingly increased both the quality and originality scores. However, my opinion and suggestions remain the same as those of Reviewer Dp9f, so I am keeping the overall rating unchanged and look forward to more details (e.g., dataset, training parameters, model weights, etc.).

One minor concern is that, to the best of my knowledge, AI for Science research suffers from a lack of real-world data. This is why distillation and synthetic data are often used, rather than professional data used in this paper. Therefore, the generalizability of the proposed method remains in doubt.

**Limitations:**

Yes.

**Paper Formatting Concerns:**

No major formatting issues were observed.

**Quality:**

3

**Strengths And Weaknesses:**

**Strengths:**

1. The paper tackles an interesting challenge - how to extend LLMs to perform effectively in domains with limited natural language corpora but rich structured expert data. Go serves as a compelling testbed with clear evaluation metrics.

2. The methodology is sound and rigorously evaluated. Dataset construction and training details are clearly documented.

**Weaknesses:**

1. While the mixed fine-tuning with structured Go expertise and general long Chain-of-Thought (CoT) reasoning data as a cold start, followed by reinforcement learning is well-executed, the overall training strategy closely resembles existing practices in LLMs.

2. The authors claim that the method can generalize to other specialized domains, yet the paper lacks concrete examples or preliminary results beyond the Go domain. While the approach is demonstrated effectively on Go, the feasibility of applying this method to other specialized domains remains speculative. The reliance on structured expert data (e.g., KataGo) and hand-crafted heuristic rules may limit scalability to less formalized domains.

---

> ### Author Rebuttal · Authors · 2025-07-27
>
> Thank you for your valuable comments! We sincerely appreciate your review, detailed comments, and valuable suggestions, and are pleased to address your concerns and provide the following clarifications. We will clarify the questions and the weaknesses in order, and we sincerely hope our responses can address your concerns.
>
> ## **Question 1: Could the authors clarify what specific aspects of the training pipeline are novel? How does this approach differ from existing fine-tuning, RLHF or domain-alignment methods?**
>
> Thank you for your insightful question! Indeed, our method design draws inspiration from works such as DeepSeek-R1, which enhance model reasoning performance through a combination of cold start and RL. However, we must emphasize the significant differences between our work and these prior approaches.
>
> In many existing works, the cold start phase **relies on distilled data** from stronger reasoning LLMs (e.g., Fin-R1[1], Med-R1[2], ReTool[3], Table-R1[4], Advancing Multimodal Reasoning via Reinforcement Learning with Cold Start[5], etc.). For tasks like Go, however, general LLMs perform poorly, making the idea of using stronger model distillation for cold start **infeasible**.
>
> To address this, we propose a novel cold start + RL approach. Instead of distilling from strong LLMs, we construct expert-level Go dataset based on heuristic rules only **(which is scalable and does not require a generative model)**, and mix it with a long COT general reasoning dataset for cold start training. Next, by adjusting the instructions, we **guide the model to perform Go tasks in the form of long COT reasoning** during the GRPO phase, successfully transferring the Go expertise learned in the cold start stage to the long COT reasoning paradigm.
>
> Thus, our propose method **combines the reasoning capabilities of large language models on general tasks with structured, non-natural language expert knowledge from out-of-distribution (OOD) tasks**, ultimately enabling general LLMs to perform high-quality natural language reasoning on specialized tasks. During the RL phase, we combine the ***reasoning ability*** learned from general reasoning data with the ***Go knowledge*** acquired from specialized Go dataset, and successfully transfer the expert knowledge into the long COT reasoning paradigm. Compared to tasks that can directly use distillation datasets or high quality natural language corpora for cold start, our method **both provides and proves the effectiveness of a novel practice for specialized domains like Go**, where NL corpora are scarce.
>
> In short, our method differs from existing training paradigms in terms of the **problem being solved**, **the data systhesizing approach for cold-start**, **the purpose of RL**, and the **specific experimental design of RL**. Therefore, our method possesses its own novelty. We hope the above clarification addresses your concerns!
>
> ## **Question 2:  Could the authors discuss concrete application areas where similar structured expert knowledge and heuristic rules could be constructed at scale? Are there any early results or experiments in these domains?**
>
> Thank you for this meaningful question. We are willing to discuss related domains to provide insights for future work. In fact, we believe that as long as the environment can **provide timely and correct feedback**, and there exists methods to **construct domain-specific data at scale**, our approach can be applied to enable general LLMs to learn reasoning tasks in specialized fields.
>
> Here is a possible idea: taking **circuit design** as an example, suppose we already have a large amount of simulation data(which indeed is easy to acquire at scale). We can then construct a domain-specific SFT dataset for circuit knowledge. By mixing such a specialized dataset with a general-domain long COT reasoning dataset for a mixed cold start. Similar to our practice on Go, we can use reasoning-oriented instructions in circuit design tasks, encouraging the model to output both reasoning processes and design results(**which can then be automatically verified for correctness in a simulation environment**). In this way, we can obtain a general model capable of both reasoning and correctly performing circuit design tasks.
>
> The core of this problem is that, for a specialized domain, large-scale data is often **structured, standardized, and non-natural language**. Our method focuses on **combining** such structured domain knowledge with the reasoning abilities acquired by the model on general reasoning tasks, enabling the model to transfer knowledge learned from specialized data into natural reasoning thoughts. Therefore, as long as a specialized task has a **sufficiently large structured dataset** and **an environment for validation**, our approach is a promising direction to consider. Given that we have already validated the effectiveness of this method on the challenging task of Go, we believe it also has potential for application in more specialized domains.
>
> For the second question (early results or experiments in these domains), we are currently conducting research in scientific areas, and experiments are still in the design phase.
>
> We hope the above clarifications address your concerns!
>
> ## **Question 3: Does the Go domain offer any general insights that can inform applications in other fields?**
>
> Thank you for this valuable question! In fact, our choice of the Go domain for experiments is **carefully considered**. First, the next-move prediction task in Go is a challenging and specialized task, with the corresponding natural language corpus for this task is essentially **non-existent** on the Internet. Therefore, Go is almost completely an **out-of-distribution (OOD) task** for both open-source and closed-source general LLMs, and can thus be viewed as a professional task requiring extensive domain expertise. Given this premise, we believe that if general-purpose reasoning models can achieve professional-level performance on the Go task, then reaching human expert levels on similar professional tasks **with scarce natural language corpus** is also a possible future achievement.
>
> Meanwhile, the Go task already has powerful policy models such as KataGo and AlphaGo, making it feasible to **obtain scalable structured domain knowledge**. Alse, existing AI models can serve as reference benchmarks for comparing model performance, allowing us to conveniently verify the performance level achieved by our models.
>
> Based on this foundation, we successfully demonstrate that our model achieves professional-level Go performance while maintaining outstanding general-purpose capabilities, which fully proves the effectiveness of our approach. Therefore, our task **provides a potential solution for other similar professional tasks with comparable characteristics (poor LLM performance, scarce natural language corpus, but scalable structured domain knowledge available)**, namely leveraging the natural generalization of reasoning paradigms in reasoning models on OOD tasks, and transferring structured domain knowledge into long CoT reasoning outputs through self-exploration.
>
> ## **Weakness 1:**
> Please refer to our clarification to question 1.
>
> ## **Weakness 2:**
> Please refer to our clarification to question 2 and 3.
>
> ## **Summary**
>
> Finally, thank you once again for your time and effort during the review process! We hope that these clarifications address your concerns about the paper, and we look forward to engaging in further discussions with you during the Discussion phase!
>
> ## **References**
> [1] Liu Z, Guo X, Lou F, et al. Fin-r1: A large language model for financial reasoning through reinforcement learning[J]. arXiv preprint arXiv:2503.16252, 2025.
>
> [2] Lai Y, Zhong J, Li M, et al. Med-r1: Reinforcement learning for generalizable medical reasoning in vision-language models[J]. arXiv preprint arXiv:2503.13939, 2025.
>
> [3] Feng J, Huang S, Qu X, et al. Retool: Reinforcement learning for strategic tool use in llms[J]. arXiv preprint arXiv:2504.11536, 2025.
>
> [4] Yang Z, Chen L, Cohan A, et al. Table-r1: Inference-time scaling for table reasoning[J]. arXiv preprint arXiv:2505.23621, 2025.
>
> [5] Wei L, Li Y, Zheng K, et al. Advancing Multimodal Reasoning via Reinforcement Learning with Cold Start[J]. arXiv preprint arXiv:2505.22334, 2025.

---

> > ### Comment · Reviewer_vfsx · 2025-08-05
> >
> > Thank you for the detailed clarification. It is very helpful. However, I still feel that some important empirical factors could significantly affect model performance but are not clearly addressed in the paper. In particular, have you investigated whether the proportion of rule-based generated data and CoT data in the mixed dataset influences the model's performance? How was the data mixing ratio determined?
> >
> > It is somewhat surprising that simply mixing rule-based expert data with general CoT reasoning data during the cold start phase can lead to effective generalization to reasoning tasks that lack natural language supervision. I appreciate the circuit design example mentioned in the rebuttal, but I am unsure how broadly applicable this is beyond that specific domain. Are there any existing studies or experimental results that support the generalizability of this approach?

---

> > > ### Author Response · Authors · 2025-08-08
> > >
> > > Dear Reviewer vfsx,
> > >
> > > Thank you once again for your detailed review and feedback! We greatly appreciate the time you've invested in reviewing our work.
> > >
> > > Given that **the discussion period is nearing its end**, we would like to check if our responses have addressed your concerns, or if there are any remaining questions to be discussed. We are very happy to continue the discussion and provide any additional information that would be helpful.
> > >
> > > Thank you again for your valuable input.
> > >
> > > Best regards,
> > >
> > > The Authors

---

> ### Author Response · Authors · 2025-08-05
>
> Dear Reviewer vfsx,
>
> Thank you once again for your thoughtful review and the time you dedicated during the review process! We would like to know your thoughts on the clarifications we provided in our response, and we are very willing to continue discussing any remaining concerns you may have.
>
> Thank you for your consideration.
>
> Best regards,
>
> The Authors

---

> ### Author Response · Authors · 2025-08-06
> **Response to Reviewer vfsx**
>
> Thank you for your thoughtful feedback! First, we are very pleased that our clarification has helped you understand our work to some extent! At the same time, we are also willing to provide further clarifications for the issues you mentioned.
>
> ## **Issue 1: Regarding Data Mixing Ratio**
>
> This is a thoughtful question. In fact, **we did conduct extensive experiments**, and the results showed that **simply mixing the two types of data with equal training epochs (1:1 ratio)** is sufficient to solve the problem. We are willing to add these experiments to the appendix as well. Below we will elaborate on the experimental details:
>
> First, we should state that in the first series of experiments, we use **7M Go data**, while using **7M reasoning data**. We compare the performance changes of the model in the RL stage on KataGo-Bench-1K when using 1 epoch and 2 epochs of Go data as cold start, while fixing the reasoning data training amount to 1 epoch.
>
> The following table shows the experimental results of the RL stage:
>
> | Training Steps | 1 Epoch Go | 2 Epoch Go|
> |-------------|---------|---------|
> | Initial Point | *63.46%* | *68.04%* |
> |RL-20Steps | 65.32% | 67.90% |
> |RL-60Steps | 71.49% | 71.83% |
> |RL-100Steps | 74.79% | 75.45% |
> |RL-200Steps | 80.68% | 79.86% |
> |RL-300Steps | 81.40% | 80.06% |
> |RL-400Steps | 83.16% | 82.54% |
> |RL-500Steps | 82.95% | 81.30% |
> |RL-600Steps | 83.06% | 83.16% |
> |RL-700Steps | 84.19% | 83.44% |
> |RL-800Steps | 85.16% | 85.26% |
> |RL-900Steps | 85.99% | 85.68% |
> |RL-1000Steps | 85.13% | 85.50% |
> |RL-1100Steps | 86.61% | 86.50% |
> |RL-1200Steps | 86.19% | 86.30% |
> |RL-1300Steps | 86.92% | 86.26% |
> |RL-1400Steps | **87.54%**| 86.71% |
>
> It can be found that compared to experiments using 1 epoch Go data for cold start, using 2 epoch shows better performance **at the initial point**, but as RL progresses, the 2 epoch Go model does not show better RL performance. Instead, **the 1 epoch model ultimately achieves a higher performance ceiling**.
>
> Additionally, we tested the performance changes on **various general tasks** when mixing different amounts of Go data with a fixed amount of reasoning data(7M) after training for 1 epoch both :
>
> | Dataset | Without Go | 500K Go | 2M Go | 4M Go | 10M Go |
> |---------|------------|---------|---------|---------|----------|
> | MATH | 97.0 | 97.1 | 96.2 | 97.4 | 97.4 |
> | AIME | 73.2 | 72.3 | 72.5 | 67.9 | 72.9 |
> | OmniMath | 65.0 | 65.1 | 66.4 | 66.3 | 66.3 |
> | GPQA_diamond | 61.9 | 61.2 | 60.1 | 59.6 | 60.6 |
>
> It can be found that mixing more Go data **does not lead to a significant impact on general reasoning capabilities**. Based on the above results, we finally chose to use **1 epoch of Go data (10M) mixed with approximately 7M long COT reasoning data** as the cold start experimental setting. We will add the above experimental results to the appendix, thank you for your question.
>
> We hope the above clarifications address your concerns!
>
> ## **Issue 2: Generalizability of the Approach**
>
> Thank you for this insightful question. To our knowledge, most attempts to apply general LLMs to specialized domains still heavily **rely on data distillation or human expert annotated data**. Our work is **the first to propose this method**, based on mixed cold start of professional data and general reasoning data, followed by RL, to achieve generalization of reasoning capability in specialized domains. Therefore, whether this method can be applied to more domains **requires follow-up work to confirm**.
>
> From our perspective, in many **AI for Science(AI4S) domains**, our method has the potential to enable general models to learn reasoning capabilities for specialized tasks from structured professional data. We are also conducting experiments on such topics, like circuit design.
>
> ## **Summary**
>
> Finally, thank you for your detailed feedback! We hope our response addresses your concerns. If you have any other concerns regarding our clarification, we would also be happy to continue engaging in discussion.

---

### Note · Authors · 2025-08-12

Dear AC/PC/Reviewers:

We sincerely thank you for your dedicated efforts in the review process! We are happy that the reviewers found our work to have "sound and rigorously evaluated methodology"(`vfsx`), "enjoyable to read"(`eZ1d`), "a promising direction"(`AjQt`), and "strong empirical claims" (`Dp9f`). We are happy to provide our following final remarks about the key points discussed.

## **Addressed Concerns:**

**1. What aspects of the training pipeline are novel?(`vfsx`(Q1) and `eZ1d`(Q2))**

Our clarification resolved reviewer `eZ1d`'s concern, and reviewer `vfsx` had no further questions.

**2. How to address 'context curse'? (`AjQt` (Q1) and `Dp9f` (Q1))**

We demonstrated 2D board rendering to reviewer `AjQt`. We referenced this demo in our response to reviewer `Dp9f` and provided the full demo further. This addressed reviewer `AjQt`'s concern.

**3. Information about data source(`AjQt` (W3), `Dp9f` (W5), `1nTB`)**

We provided explanations and committed to adding the information to the appendix.

## **Remaining Concerns:**

We **did not have the opportunity to continue further discussion**  regarding our clarifications on these questions.

**1. LLMs struggle with long sequential data, unconvinced by our approach(reviewer `Dp9f`)**

We have provided **extensive evidence including game records, model responses, Elo ratings, e.g.**. Thus, we must kindly argue that this perspective is biased, as experiments demonstrate LLMs' ability to solve Go tasks.

Additionally, we will open an online platform for Go players to experience LoGos, and commit to open-sourcing our datasets, models, and benchmark.

**2. Data Mixing Ratio(Reviewer `vfsx`)**

We provided experimental results on RL performance and general domain performance under different mixing ratios.

**3. Generalizability of the Approach(Reviewer `vfsx` (Q2&3))**

Our method could be applied to some AI4S areas, such as circuit design. Since we are the first to propose this method, whether it can be applied to more domains requires follow-up work.

**4. Explanation for choosing text encoding(Reviewer `Dp9f`)**

Our 2D board rendering effectively addresses challenges in longer game states. Since text encoding aligns with LLMs' input paradigm, and Logos already achieves pro-level performance, this approach is viable.

## **Conclusion**

We believe we have provided detailed responses to each reviewer's concerns. Once again, thanks for your feedback and engagement!

Best regards,

The Authors

---

### Decision · Program_Chairs · 2025-09-17

**Decision:**

Accept (poster)

**Comment:**

This paper presents LoGos, a general-purpose LLM adapted to the game of Go through a mixed cold-start strategy and reinforcement learning. While building on existing techniques, LoGos achieves professional-level Go performance while preserving general reasoning abilities—an interesting and potentially valuable result for the community.

Notably, the authors have committed to open-sourcing their dataset, models, and benchmark in the rebuttal, which would significantly enhance the impact and reproducibility of this work. Given the borderline scores, I believe acceptance is justified only under the condition that the authors fulfill this open-source commitment.

Therefore, I recommend acceptance contingent upon the public release of these resources as promised. I sincerely hope the authors follow through on this commitment to ensure the broader community can benefit from their work.